# Next generation of Bluelink ocean reanalysis with multiscale data assimilation: BRAN2020

Matthew A. Chamberlain[1], Peter. R. Oke[1], Russell A. S. Fiedler[1], Helen M. Beggs[2], Gary B. Brassington[2], and Prasanth Divakaran[2]

[1]CSIRO Oceans and Atmosphere, Hobart, TAS, Australia
[2]Bureau of Meteorology, Australia

**Correspondence:** Matthew A. Chamberlain (matthew.chamberlain@csiro.au)

**Abstract.** BRAN2020 is an ocean reanalysis that combines observations with an eddy-resolving, near-global ocean general circulation model, to produce a four-dimensional estimate of the ocean state. The data assimilation system employed is ensemble optimal interpolation, implemented with a new multiscale approach that constrains the broad-scale ocean properties and the mesoscale circulation in two steps. There is a separation in the scales that are corrected in the two steps; the high-resolution step corrects the mesoscale dynamics the same way as previous versions of BRAN, while the extra coarse step is effective at correcting biases that develop at large scales. The reanalysis currently spans January 1993 to December 2019, and assimilates observations of *in situ* temperature and salinity, as well as satellite sea-level anomaly and sea surface temperature. BRAN2020 is planned to be updated to within months of real time after this initial release while it is the most current version. Reanalysed fields from BRAN2020 generally show much closer agreement to observations than all previous versions with mis-fits between reanalysed and observed fields reduced by over 30% for some variables, for subsurface temperature and salinity in particular. The BRAN2020 dataset is comprised of daily-averaged fields of temperature, salinity, velocity, mixed-layer depth, and sea-level. Reanalysed fields realistically represent all of the major current systems within 75°S and 75°N, excluding processes relating to sea ice, but including boundary currents, equatorial circulation, Southern Ocean variability, and mesoscale eddies. BRAN2020 is publicly-available at https://doi.org/10.25914/6009627c7af03 (Chamberlain et al., 2021b) and is intended for use by the research community.

## 1 Introduction

An ocean reanalysis combines ocean observations with a model to produce gridded estimates of the ocean state. The dataset presented here is the 2020 version of the Bluelink ReANalysis (BRAN2020; 2020 refers to the year in which the configuration was finalised). Previous versions of BRAN include BRAN1 (Oke et al., 2005), BRAN1p5 (Oke et al., 2008), BRAN2 (Schiller et al., 2008b), BRAN3 (Oke et al., 2013b), BRAN2015 (Oke et al., 2018, when the naming convention changed), and BRAN2016 (same configuration as BRAN2015 but initialised in the 1990s). Overall, the accuracy of BRAN2020 is much better than all previously presented versions.

BRAN has been developed as part of the Bluelink Project, an Australian partnership between CSIRO, the Bureau of Meteorology and the Department of Defence. The development of BRAN has supported the operational oceanographic forecasting and focused on the dynamics of the upper ocean around Australia and surrounding regions. The long, high-resolution ocean reanalyses in the various BRAN products have found various operational and research applications (see Schiller et al., 2020, for an overview). For instance, previously BRAN output has been used to support the study of hard to observe features such as boundary currents in the Asian-Australia region (Schiller et al., 2008a) and zonal currents across the Indian Ocean (Divakaran and Brassington, 2011), anomalous temperature events in the Coral Sea (Schiller et al., 2009) and along New South Wales (Oke and Griffin, 2011). The reanalyses also provide boundary conditions to regional models (e.g., the Great Barrier Reef, Steven et al., 2019).

There are two main differences between BRAN2020, and all previous versions of reanalyses in this series. The first difference is the adoption of a multiscale data assimilation technique (Chamberlain et al., 2021a) that constrains the broad-scale features of the ocean and the mesoscale features separately in two steps. Chamberlain et al. (2021a) showed that this delivers an improvement in accuracy – measured by the mis-fit to assimilated and withheld observations – of up to 35%. The same improvement is realised here, in BRAN2020. Multiscale data assimilation has also been applied to regional and basin scale ocean models (Li et al., 2015; Carrier et al., 2019; Tissier et al., 2019) which see similar improvements in overall performance. The second difference is the assimilation of a larger observational dataset that includes observations from marine mammals, moorings, and ship-borne surveys that were not assimilated into previous versions of BRAN. Previous versions only assimilated observations from the conventional observation platforms (Argo; eXpendable BathyThermographs, XBTs; ship-borne surveys; satellite sea-level anomaly, SLA; and satellite sea-surface temperature, SST). The new sources of data offer greater spatial coverage, with fewer gaps.

This paper is intended to provide a description of how BRAN2020 data are generated and the nature of model-observation differences, with sufficient detail that a user can assess whether the data are suitable for their intended application. The structure of the paper is as follows. Section 2 describes the reanalysis cycle, ocean model, data assimilation system, and observations assimilated. The discussion of the data assimilation system includes a summary of the multiscale data assimilation approach, with technical details of the assimilation reported in Appendix A. Section 3 describes the results, including an evaluation of the quality of reanalysed fields and inter-comparisons with previous BRAN experiments. Section 4 describes the output available, access and structure of files, and, Section 5 discusses potential uses of the BRAN2020 dataset and concludes.

## 2 Ocean Reanalysis Methods

In this section we describe the procedures followed to generate BRAN2020. We provide a description of the analysis cycle, how the model and observations are combined; and descriptions of the ocean model, the data assimilation system, and the observations assimilated.

## 2.1 Analysis cycle

The initial condition for the ocean state at the beginning of BRAN2020 is interpolated from climatology, using the 2013 version of the World Ocean Atlas (WOA13; Zweng et al., 2013; Locarnini et al., 2013). The first three months of the integration (October-December 1992; comprising 30 analysis cycles) are not included in the publicly available dataset. There are large adjustments in the first several analysis cycles as the reanalysed fields approach the observed ocean state from initial conditions, as seen in reductions in averaged model-observation differences with each analysis cycle. After 30 cycles these differences have

stabilised.

The analysis cycle involves alternate running of the ocean model and data assimilation system. For BRAN2020, observations are assimilated every three days, as used in previous versions (Oke et al., 2018). The instantaneous fields for salinity, sea level and horizontal velocity, and daily-averaged fields of temperature (for the day immediately preceding the analysis time) are used as the background field for the data assimilation step. Daily-averaged temperature is used as the background field instead

of instantaneous fields to avoid systematic, regional biases due to the diurnal cycle of surface temperatures. Temperature fields are also converted from potential temperature (used by the model) to *in situ* temperature to match the observations to be assimilated. This conversion has no impact at the surface, but changes the temperature by about 0.2°C at 2000 m depth.

The differences between the background field and observations are calculated by interpolating the model fields to the observation locations, yielding a vector of background innovations. Background innovations represent the differences between the

model and observations before assimilation. The data assimilation system is used to calculate the analysis field (fully described in Section 2.3). The differences between the analysis field and observations are also calculated, yielding a vector of analysis innovations, also referred to as "residuals". Analysis innovations represent the differences between the model and observations after assimilation. The differences between the analysis field and the background fields are here referred to as the increments. Increments include values for all variables of the model state: temperature, salinity, horizontal velocities, and sea-level; for

all model grid points. The increments are added to the instantaneous fields at the end of the previous ocean model step, or at analysis time, for all variables, to reinitialise the model for the next analysis cycle. For salinity, horizontal velocities, and sea-level, this means that the model is initialised with the analysis fields. But because temperature is treated slightly differently, as described above, the initialised temperature does not precisely match the analysis fields.

This analysis cycle is repeated every three days for the period of January 1993 to December 2019. Background and analysis

innovations, for each cycle, are used to quantify the accuracy of the reanalysis in Section 3. Daily-averaged fields of temperature, salinity, velocity, sea-level and mixed layer depth are stored for each day of integration. The mixed layer depth here is the depth over which the buoyancy exceeds a threshold of 0.0003 m/s$^2$, as described by Griffies (2012). These daily-averaged fields comprise the BRAN2020 product that is presented here.

## 2.2 Ocean model

The model configuration used here is the Ocean Forecasting Australian Model, version 3 (OFAM3). A comprehensive description of OFAM3 is presented by Oke et al. (2013a). Briefly, OFAM3 is a near-global, eddy-resolving, z$^*$ configuration of the

Modular Ocean Model (version 5, Griffies, 2012), developed principally for the purpose of reanalyses and forecasting upper ocean conditions across the globe, excluding the polar regions. The model grid has 0.1° grid resolution between 75°S and 75°N, with 5-m vertical resolution down to 40-m depth and 10-m vertical resolution to 200-m depth. OFAM3 is forced with atmospheric conditions from JRA-55 (Kobayashi et al., 2015; Tsujino et al., 2020) using bulk formulas to calculate surface fluxes of heat, freshwater and momentum. The model is ocean-only and does not include sea ice. Forcing fields are masked for sea ice extent from the atmospheric reanalysis, to avoid unrealistic surface fluxes that would result from bulk formulas with open water exposed to cold winter conditions at high-latitudes. Where sea ice is present, values in the atmospheric reanalysis fields are replaced with values expected below sea ice; for instance, winds and downward shortwave radiation are set to zero, while downward thermal radiation is set to the thermal radiation at freezing temperature.

Surface salinity is restored to a monthly climatology, using WOA13 (Zweng et al., 2013), with a restoring time-scale of 14 days. Temperature and salinity below 2000 m are also relaxed towards WOA13 climatology (Zweng et al., 2013; Locarnini et al., 2013), with a restoring time-scale of 1 year. The configuration of OFAM3 used here employs the K-epsilon scheme (Rodi, 1987) to calculate unresolved turbulent mixing in the upper ocean.

## 2.3 Ocean data assimilation system

### 2.3.1 Ensemble optimal interpolation

The assimilation system used to constrain BRAN2020 is EnKF-C (Sakov, 2014), implemented in the Ensemble Optimal Interpolation (EnOI) mode. Technical details, specific to the EnKF-C software including all parameters and settings, are presented in Appendix A. For EnOI, the state of the ocean model, $\boldsymbol{w}$ (comprised of temperature, salinity, horizontal velocity, and sea-level), is updated by projecting background innovations onto an ensemble of model anomalies, $\mathbf{A}$, and calculating weightings, $\boldsymbol{c}$, for each member. The weights vary horizontally and are recalculated for each analysis cycle. The increment, used to update the ocean state, is a weighted sum of ensemble members:

$$\boldsymbol{w}^a = \boldsymbol{w}^b + \boldsymbol{w}^{inc} \tag{1}$$

$$= \boldsymbol{w}^b + \sum_{i=1}^{n} \boldsymbol{c}_i(x,y).\mathbf{A}_i(x,y,z), \tag{2}$$

where superscripts $a$, $b$, and $inc$ denote analysis, background, and increment fields, respectively. Subscripts $i$ denote the $ith$ ensemble member, $n$ is the ensemble size, and $x$, $y$, and $z$ are dimensions in space.

The weights, in Eq. (2), depend on many factors, including the assumed relative size of the observation error variance and background field error variance. These weights also depend on the correlations between the background innovations and the anomalies in the ensemble members. If, for example, there is no combination of ensemble members that can "fit" the background innovations, then the ensemble can be considered rank-deficient - lacking sufficient degrees of freedom to fit the data. In that case, the analyses won't match the observations well. The details of the assumed observation errors, and the details of the ensembles can have significant impact on the performance of the data assimilation and the overall reanalysis.

### 2.3.2 Multiscale data assimilation

A new feature in BRAN2020 is the adoption of multiscale data assimilation which is particularly effective at reducing the bias
in the model in the subsurface. The method is described briefly here, for details please see Chamberlain et al. (2021a).

In BRAN2020, two ensembles are constructed based on anomalies derived from two previous experiments using free-running models with no data assimilation. A strength of this method is that it is multivariate. This means that observations of one variable can be used to calculate increments for other variables. For example, the assimilation of SLA produces an update to velocity, even when velocity observations are not assimilated. To achieve this, the ensemble fields are used to characterise
the statistical relationships - the covariance - between each model variable. In practice, the covariability of different variables at different locations are represented by the ensemble members. By constructing the ensembles in this way, we're making an assumption that the background field errors can be represented by the sample of anomalies in these ensembles. Moreover, here with multiscale data assimilation, we're assuming that there are background field errors on both broad scales and mesoscales.

For BRAN2020, the data assimilation is implemented in two steps – the coarse- and high-resolution assimilation – using
two different ensembles at the different resolutions. The coarse-resolution ensemble is generated by a one-degree resolution global ocean model (the ocean component of the Australian Community Climate and Earth System Simulator, ACCESS; Bi et al., 2013), using a configuration like Kiss et al. (2019) but using the sea ice component with the Modular Ocean Model (Sea Ice Simulator, Delworth et al., 2006). The coarse ensemble is constructed with monthly climatological anomalies, with one member constructed for each month of a free-running 40-year model simulation. This ensemble is intended to represent
the potential broad-scale background field errors. The high-resolution ensemble is generated by a OFAM3 spinup (Oke et al., 2013a). The high-resolution ensemble contains seasonal-scale anomalies, constructed by centred differences of 3-day from 3-month averages, with one member constructed for each month of a 12-year model simulation, without data assimilation. This ensemble is intended to represent the potential mesoscale background field errors.

The first step, the coarse-resolution assimilation, is intended to correct the broad scales of the ocean state. To calculate
increments for the coarse assimilation step, the background field on the high-resolution grid $\boldsymbol{w}^b_{high-res}$, is first spatially-averaged onto a one-degree grid, yielding $\boldsymbol{w}^b_{coarse}$. The spatially-averaged background field is compared to observations, yielding the background innovations for the coarse-resolution assimilation step. EnOI is used to project these innovations onto the 480-member coarse-resolution ensemble of anomalies to produce a coarse-increment $\boldsymbol{w}^{inc}_{coarse}$. The increments are interpolated back to the 0.1° grid, yielding $\boldsymbol{w}^{inc}_{inter-coarse}$, and added to $\boldsymbol{w}^b_{high-res}$. This gives an updated background field on
the high-resolution grid $\boldsymbol{w}^{updated-b}_{high-res}$:

$$\boldsymbol{w}^{updated-b}_{high-res} = \boldsymbol{w}^b_{high-res} + \boldsymbol{w}^{inc}_{inter-coarse}, \tag{3}$$

ready for the second step of the assimilation. We find that this first step largely constrains the broad scales of the ocean state, without impacting the mesoscales.

The second step, the high-resolution assimilation, is intended to correct the mesoscale structures of the ocean state and uses
the same configuration as recent versions of BRAN (Oke et al., 2018). The updated background field on the high-resolution grid, $\boldsymbol{w}^{updated-b}_{high-res}$, is compared to observations (the same observations assimilated in the first step) yielding the background innova-

tions for the high-resolution assimilation step. EnOI is used to project these innovations onto a 144-member high-resolution ensemble of anomalies to produce a high-resolution increment $\boldsymbol{w}_{high-res}^{inc}$. These increments are added to the respective variables in the model's restart files, and the model is integrated forwards in time. We find that this step largely constrains the mesoscales of the ocean, mainly adjusting the locations and properties of mesoscale eddies, and adjusting their strength, location, and properties.

As demonstrated in Chamberlain et al. (2021a), even though the same observations are used, there is a clear separation of scales in the corrections from each assimilation step. The length scale of the corrections are determined when the assimilation system projects the observation-model differences onto ensemble members. Here, the ensemble members of the coarse step are climatological anomalies with scales much longer than the high-resolution ensemble with seasonal-scale anomalies.

## 2.4 Ocean observations

BRAN2020 assimilates observations from satellite SST, satellite SLA, and *in situ* temperature and salinity. BRAN2020 uses updated sources of observational data relative to previous versions of BRAN. Figure 1 is a "Gannt" chart summarising the temporal extent of various observation products that are assimilated into BRAN2020.

For most years, the European Space Agency SST Climate Change Initiative (CCI-SST, Merchant et al., 2019) is the source of Advanced Very High Resolution Radiometer (AVHRR, Embury et al., 2019a) and Along Track Scanning Radiometer (ATSR, Embury et al., 2019b) SST observations. Data from CCI-SST include updated calibrations and quality controls, spanning 1981 to 2016; "best quality" (level 5), day- and night-time temperatures at depth (0.2 m) data are assimilated into BRAN. Note, these SST data are compared to a daily averaged background; the difference between instantaneous and daily averaged surface temperature is small in BRAN since the 5-m resolution is greater than the typical diurnal variation ($\sim$1 m). From year 2017 onwards, AVHRR data (again, day and night time observations) are sourced from NAVO (Naval Oceanographic Office, 2014a, b, c, d). Night-time microwave SST data from AMSRE (Gentemann et al., 2010) and AMSR2 (retrieved from the Japan Aerospace Exploration Agency, ftp://ftp.eorc.jaxa.jp/AMSR2/GDS2.0/) are assimilated from 2009, where available. Microwave SST data are not as accurate as infrared observations, but microwave SST data are not as affected by clouds (in the absence of rain), thus providing a good complement to infrared-based observations. Night-time SST data from Visible Infrared Imaging Radiometer Suite (VIIRS, Petrenko et al., 2014; NOAA Office of Satellite and Product Operations, 2019) aboard Suomi National Polar Partnership (NPP) and NOAA-20 satellites, are included from 2012 and 2018 respectively. The VIIRS SST data have somewhat superior spatial coverage to AVHRR SST, due to the higher spatial resolution (0.75 km to 1.5 km for VIIRS compared with 9 km to 30 km).

Observations of SLA are assimilated into BRAN2020, using along-track satellite altimeter data from various satellite platforms that have been available since the 1990s (Fig. 1). SLA data are sourced from the Radar Altimeter Database System (RADS Ver. 4, retrieved from http://rads.tudelft.nl, Scharroo et al., 2013), and include corrections for astronomical tides and inverted barometer effects. SLAs in the model are referenced to a mean sea-level, calculated from an 18-year mean of an OFAM3 experiment without data assimilation (the same experiment that was used to construct the high-resolution ensemble).

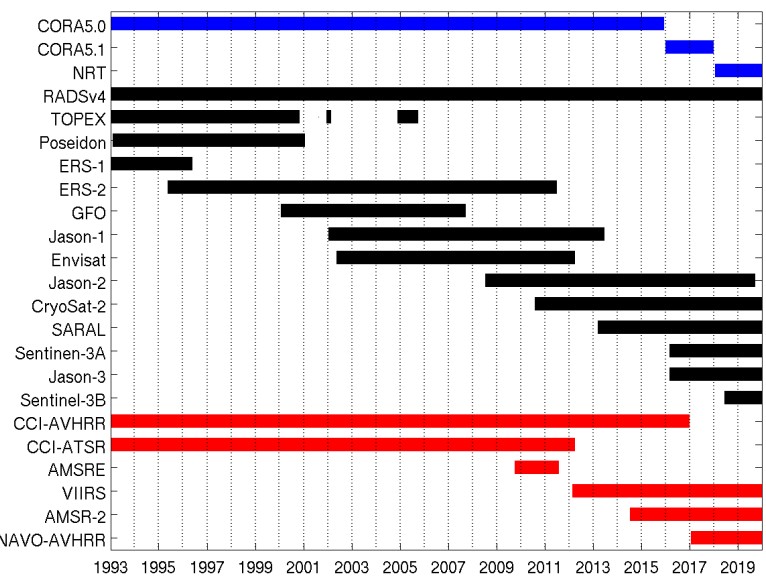

**Figure 1.** Gantt chart showing when data from different types and sources are assimilated into BRAN2020. Altimeter data are accessed from RADSv4, accessing data from 13 different satellite altimeters (black bars). SST data are accessed from five different sources (red bars), including data from individual satellites (VIIRS, AMSRE, AMSR-2) and from datasets that have combined AVHRR and ATSR data from multiple satellites (CCI-AVHRR, CCI-ASTR and NAVO-AVHRR). In situ data are accessed from three sources - two from Copernicus Marine Environment Monitoring Service (CMEMS; Coriolis Ocean dataset for Re-Analysis, CORA5.0 and CORA5.1) and a near-real-time dataset maintained by the Bureau of Meteorology.

Consequently, the mean barotropic currents of BRAN2020 will be very similar to those of this free-running experiment. SLA observations are not assimilated in water less than 200 m.

*In situ* observations of temperature and salinity are assimilated into BRAN2020, using measurements from Argo floats, surface drifting buoys, XBT (temperature only), sea mammals, moorings, as well as ship-borne surveys, and underway systems. These data are sourced from the Coriolis Ocean Dataset for ReAnalysis (CORA, versions 5.0 and 5.1, Cabanes et al., 2013;

Tanguy et al., 2019) and from a near-real time (NRT) database maintained at the Australian Bureau of Meteorology. CORA databases include data that have been processed with delayed-mode quality control. CORA observation types with higher uncertainties, such as XBT and sea mammals, are assigned larger errors in types TEM2 and SAL2 in Table 1. The NRT database, used from 2018 onwards in BRAN2020 (Fig. 1), includes data sourced from the Global Telecommunications System and the Argo Global Data Assembly Centres (Argo, 2021). These data are subject to some NRT quality control and are

duplicate-checked to exclude multiple versions of the same observations (Brassington et al., 2012).

For every assimilated observation, an explicit observation error estimate is required. Specifically, the standard deviation of observation errors need to be estimated. Observations are assumed to be unbiased. Observation errors include measurement

**Table 1.** Summary of observations and errors used when assimilated into BRAN. TEM/SAL includes Argo, CTD, moorings etc.; TEM2/SAL2 includes XBT, marine mammals.

| Observation Type and Platform | Assumed Standard Deviation of Observation Errors |
|---|---|
| SST (AVHRR) | 0.1°C |
| SST (ATSR) | 0.1°C |
| SST (AMSRE) | 0.5°C |
| SST (AMSR2) | 0.45°C |
| SST (VIIRS) | 0.15°C |
| SLA | 0.05 m |
| SLA (Geosat) | 0.07 m |
| TEM | 0.5°C |
| TEM2 | 1°C |
| SAL | 0.075 psu |
| SAL2 | 0.15 psu |

error and representation error (e.g., Oke and Sakov, 2008). Because observation errors (especially representation errors) are relatively poorly known, the estimates used for BRAN2020 were tuned somewhat, to improve the performance based on ocean reanalyses simulating recent years. Table 1 summarises observation classes and the assumed standard deviations of each observation type. As is common in data assimilation problems, it is possible to force the model ocean state to closely match observation data, but this can lead to overfitting and may cause unrealistic artefacts in the model that can degrade the quality of the reanalysis. As a result, assumed observation errors tend to be larger than some might expect. The one exception to this here, is the assumed estimate of AVHRR SST error (Table 1). For BRAN2020, we assume that the standard deviation of the AVHRR SST error is 0.1°C. In retrospect, based on the recent results presented by Merchant et al. (2019, their Fig. 6) and the assessment in Section 3, we think that the BRAN2020 performance may have been better for upper-ocean temperature if we'd assumed a larger error for AVHRR SST - perhaps with a value between 0.2 and 0.4°C. We plan to assess this for future versions of BRAN in the future. This is discussed further in Section 3.1

The observation errors actually used by EnKF-C in the data assimilation calculations are influenced by a process to build "super-observations." This is a preprocessing step applied at the start of each data assimilation cycle to combine observations of the same type and at approximately the same locations and times (in most cases, within the same model grid point). The error variance of a super-observation is calculated from the inverse of the sum of inverse variances of the individual observations used to construct the super-observation (Sakov, 2014). As a result, the error variance of each super-observation is often smaller than the assumed observation error variances of the individual observations. The practice of building super-observations is common, reducing the computational cost of the assimilation step, and as a conservative measure, eliminating some anomalous observations that may represent fluctuations not represented by the model (e.g., sub-gridscale features).

## 3 Evaluation and Assessment of Ocean Reanalysis

To demonstrate the quality of the BRAN2020 dataset, we present two classes of metrics. First, we present the model-observation differences computed during the assimilation steps - specifically looking at the background and analysis innovations, or model-observation misfits. Second, we compare daily-averaged reanalysed fields (the fields released as part of the BRAN2020 dataset) with daily observations, observation-based indices, and observation-based properties.

### 3.1 Analysis of innovations

Here, we report statistics of the background and analysis innovations, before and after assimilation. This provides insight into how closely BRAN2020 analyses "fit" the assimilated observations (using the analysis innovations), and how faithfully the model integrates analyses forward in time (using the background innovations). Comparison of the analysis and background innovations can help identify potential cases of over-fitting. Ideally, we hope for small analysis innovations and small background innovations. If analysis innovations are small, but background innovations are large, we might suspect over-fitting.

The mean absolute deviations of the analysis and background innovations are presented in Table 2 and 3, respectively. BRAN2020 statistics are presented for decadally- and globally-averaged absolute innovations for SST, SLA, and for sub-surface temperature and salinity in different depth ranges. The decade averages are presented for the 1990s, 2000s, and 2010s, to show how the quality of the reanalysed fields changes with time, as the observing system changes (Fig. 1). For comparison, the same statistics are included for BRAN2016. Background innovations of both BRAN2020 and BRAN2016 are after 3 days of integration from the previous analysis. Note, the values reported for each version are with respect to the observation products assimilated in that version; most data are common to both versions, though as noted, BRAN2020 contains extra data not included in BRAN2016 such as sea mammals and moorings. However, the BRAN2020 results here are consistent with those reported in Chamberlain et al. (2021a) which tested a multiscale reanalysis configuration using the same observations as BRAN2016. For almost all metrics reported in Table 2 and 3, BRAN2020 fields are in better agreement with observations than BRAN2016. The only exception to this is temperature in the top 50 m in the 2000s.

The greatest improvement in BRAN2020, compared to BRAN2016, is in sub-surface temperature and salinity fields (Table 2 and 3). For depths greater than 500 m, the analysis and background innovations for both temperature and salinity are of the scale of 30% smaller in BRAN2020 than they were in BRAN2016. In the decade of the 2000s, between 50 and 500 m depth, the analysis and background innovations are 20-40% smaller in BRAN2020 for both temperature and salinity. For depths shallower than 50 m in the 2000s, the analysis and background innovations for temperature are 11-14% greater in BRAN2020 than they were in BRAN2016; and for salinity, they are 3-17% smaller in BRAN2020.

BRAN2020 also demonstrates modest improvements relative to BRAN2016 in the innovations of the surface metrics. In the 2000s, the analysis and background innovations of SST are reduced by 8-23%;for SLA, they are reduced by 3-6%. These small changes are consistent with the results of Chamberlain et al. (2021a) which found improvments in the subsurface and minimal change in surface metrics assicated with the implementation of multiscale data assimilation. Based on this we conclude the improvements obtained with BRAN2020 are associated with the updated observation products.

**Table 2.** Decadal averages of mean absolute values of analysis innovations for BRAN2020 and BRAN2016. These statistics provide an indication of how closely BRAN analyses "fit" the assimilated observations.

|  | BRAN2020 | | | BRAN2016 | | |
|---|---|---|---|---|---|---|
|  | 1990s | 2000s | 2010s | 1990s | 2000s | 2010s |
| SST (°C) | 0.19 | 0.17 | 0.13 | 0.23 | 0.22 | 0.16 |
| SLA (cm) | 3.0 | 3.2 | 2.9 | 3.3 | 3.4 | 3.1 |
| Subsurface temperature (°C) | | | | | | |
| (all depths) | 0.44 | 0.35 | 0.28 | 0.50 | 0.46 | 0.39 |
| (0-50 m) | 0.50 | 0.39 | 0.29 | 0.47 | 0.35 | 0.29 |
| (50-500 m) | 0.43 | 0.37 | 0.32 | 0.52 | 0.53 | 0.45 |
| (500+ m) | 0.26 | 0.25 | 0.19 | 0.40 | 0.37 | 0.33 |
| Subsurface salinity (psu) | | | | | | |
| (all depths) | 0.085 | 0.060 | 0.046 | 0.161 | 0.090 | 0.071 |
| (0-50 m) | 0.122 | 0.086 | 0.059 | 0.198 | 0.103 | 0.072 |
| (50-500 m) | 0.069 | 0.053 | 0.047 | 0.132 | 0.092 | 0.075 |
| (500+ m) | 0.044 | 0.041 | 0.035 | 0.074 | 0.063 | 0.061 |

**Table 3.** Decadal averages of mean absolute values of background innovations for BRAN2020 and BRAN2016. These statistics provide an indication of how well the model integrates BRAN analyses forward in time.

|  | BRAN2020 | | | BRAN2016 | | |
|---|---|---|---|---|---|---|
|  | 1990s | 2000s | 2010s | 1990s | 2000s | 2010s |
| SST (°C) | 0.38 | 0.36 | 0.31 | 0.46 | 0.39 | 0.32 |
| SLA (cm) | 5.3 | 5.4 | 5.1 | 5.8 | 5.6 | 5.2 |
| Subsurface temperature (°C) | | | | | | |
| (all depths) | 0.68 | 0.56 | 0.45 | 0.81 | 0.66 | 0.56 |
| (0-50 m) | 0.73 | 0.57 | 0.42 | 0.75 | 0.50 | 0.43 |
| (50-500 m) | 0.72 | 0.65 | 0.57 | 0.87 | 0.79 | 0.69 |
| (500+ m) | 0.33 | 0.30 | 0.23 | 0.48 | 0.42 | 0.37 |
| Subsurface salinity (psu) | | | | | | |
| (all depths) | 0.148 | 0.105 | 0.082 | 0.273 | 0.131 | 0.102 |
| (0-50 m) | 0.229 | 0.175 | 0.129 | 0.349 | 0.181 | 0.134 |
| (50-500 m) | 0.118 | 0.094 | 0.085 | 0.207 | 0.132 | 0.110 |
| (500+ m) | 0.051 | 0.046 | 0.038 | 0.084 | 0.068 | 0.064 |

In addition to the improvements in mean absolute differences shown here, BRAN2020 also reduces biases in the ocean model, or the mean values of the innovations. The sub-surface fields were known to have relatively large biases in earlier versions of BRAN (e.g., Oke et al., 2013b). These seem to be mostly eliminated with the use of the multiscale data assimilation. For instance, the average of the background observation-minus-model innovations for all subsurface temperature and salinity in the 2010s of BRAN2016 are -0.185°C and +0.00771 psu, respectively; in BRAN2020 these are reduced to -0.0456°C and
+0.00196 psu. These global values can mask regional structures with significant biases that average out. However, even these regional biases are mostly eliminated with multiscale data assimilation (see Fig. 6 of Chamberlain et al., 2021a). As discussed by Chamberlain et al. (2021a), the two assimilation steps correct the ocean state at separate scales. The fine scale corrects mesoscale features (as done in BRAN2016), and the extra coarse step corrects large-scale biases.

There is a slight degradation in temperature, shallower than 50 m, that we think is an indication that the BRAN2020 analyses
are over-fitting SST. As noted in Section 2.4, we think that a better result may have been achieved if we assumed a larger observation error for AVHRR SST for BRAN2020 (Table 1). In short tests of ~20 cycles, the BRAN system evaluated the impact of increasing the observation error of AVHRR SST from 0.1 to 0.3°C. The results of mean absolute analysis innovations for subsurface temperatures, 0 - 50 m, decreased ~3% with the larger SST error, and smaller reductions of ~0.5% in forecast innovations, explaining part of the differences between BRAN2020 and BRAN2016 in Tables 2 and 3. Note again, there are
more observations assimilated and assessed in the BRAN2020 relative to BRAN2016. We note that VIIRS SST is reported as higher in spatial resolution to AVHRR SST and generally agrees closer to drifting buoy SST (Minnett et al., 2020; O'Carroll et al., 2019, their Fig. 10), although this is not reflected in our error estimates in Table 1 – and so we surmise that post-2012 (when VIIRS data are assimilated) the impact of this slight over-fitting to AVHRR is much less. Note also, ATSR has less error relative to AVHRR, as assessed with drifting buoys (Merchant et al., 2019, their Fig. 6), so the assumed ATSR error of 0.1 °C
(Table 1) is a reasonable.

To better understand whether BRAN2020 analyses over-fit any of the assimilated observations, we present differences between the mean absolute values of background and analysis innovations for BRAN2020 and BRAN2016 in Table 4. Comparison between BRAN2016 and BRAN2020 (in the 2000s, columns 3 and 6 of Table 4), suggests that the error growth between analysis cycles during the 2000s is faster for BRAN2020. This may be because either BRAN2020 over-fits some observa-
tions, or because BRAN2016 under-fits some observations - or perhaps both. We can't be sure of which explanation is true. But as noted, we think that BRAN2020 over-fits AVHRR SST; and we are quite sure that BRAN2016 under-fit sub-surface temperature and salinity below 50 m, and particularly below 500 m. These results demonstrate one of the main challenges of performing these big reanalysis experiments. There are many factors to "tune" in these systems, and there are competing goals of "fitting" data as close as possible, but not over-fitting. It looks like for BRAN2020, the configuration is achieving a good
balance - fitting the data much more closely than previous versions across most of the ocean state, but perhaps overfitting some variables in some places.

Time series of the mean absolute values of the background and analysis innovations are presented for SLA, SST, temperature (all depths), and salinity (all depths) in Fig. 2. These are the same metrics used to prepare Tables 2, 3, and 4. SLA innovations (Fig. 2a) fluctuate modestly in time, with a few step-changes when the composition of satellite altimeters changed (Fig. 1). For

**Table 4.** Decadal averages of differences between the mean absolute values of background and analysis innovations for BRAN2020 and BRAN2016. These statistics provide an indication of how the errors grow between each analysis cycle. Fast error growth may indicate that analyses are over-fit, or that the observing system is too sparse to adequately constrain the reanalysis.

| | BRAN2020 | | | BRAN2016 | | |
|---|---|---|---|---|---|---|
| | 1990s | 2000s | 2010s | 1990s | 2000s | 2010s |
| SST (°C) | 0.19 | 0.19 | 0.18 | 0.23 | 0.17 | 0.16 |
| SLA (cm) | 2.3 | 2.2 | 2.2 | 2.5 | 2.2 | 2.1 |
| Subsurface temperature (°C) | | | | | | |
| (all depths) | 0.24 | 0.21 | 0.17 | 0.31 | 0.20 | 0.17 |
| (0-50 m) | 0.22 | 0.18 | 0.13 | 0.28 | 0.15 | 0.14 |
| (50-500 m) | 0.29 | 0.28 | 0.25 | 0.35 | 0.26 | 0.24 |
| (500+ m) | 0.07 | 0.05 | 0.04 | 0.08 | 0.05 | 0.04 |
| Subsurface salinity (psu) | | | | | | |
| (all depths) | 0.063 | 0.046 | 0.036 | 0.113 | 0.041 | 0.031 |
| (0-50 m) | 0.107 | 0.089 | 0.071 | 0.152 | 0.078 | 0.062 |
| (50-500 m) | 0.048 | 0.041 | 0.038 | 0.076 | 0.040 | 0.035 |
| (500+ m) | 0.007 | 0.005 | 0.003 | 0.011 | 0.005 | 0.004 |

example, the innovations increase when data from the Topex/Poseidon satellites become unavailable at the end of the 1990s, and they reduce again when data from the Jason/Envisat satellites are available after 2002.

Over the course of BRAN2020, there is a gradual decrease in SST innovations (Fig. 2b), as the number of satellites increased and sensors improved (O'Carroll et al., 2019). As noted in Fig. 1, the main source of SST observations for BRAN2020 is CCI-AVHRR. But BRAN2020 also assimilated microwave SST from 2009, and VIIRS SST from 2012. There is no clear step change when microwave SST is used, but there is a clear reduction in both analysis and background innovations when VIIRS SST data are assimilated from 2012. VIIRS SST data have been observed to generally have smaller standard deviations when compared with drifting buoy SSTs than either AVHRR or AMSR2 SST products (O'Carroll et al., 2019; Helen Beggs, pers. comm.).

The constraint of sub-surface temperature and salinity improves consistently over the course of BRAN2020 (Fig. 2c,d). This is particularly true over the decade of the 2000s directly associated with the expansion of the Argo float array that provides systematic observations of sub-surface ocean properties with near-global coverage (Roemmich et al., 2019). Prior to the Argo array, *in situ* data were scattered in space and sporadic in time. Not only do more *in situ* observations improve analyses - with smaller analysis innovations - but they also result in better quality "forecasts", with smaller background innovations. Another prominent feature of Fig. 2c,d, is a seasonal cycle in the global averaged innovations from the beginning of BRAN2020, until about 2008. The seasonal cycle in innovations largely disappears once observations from the global Argo array nominally reached its target density in about 2008 (Wong et al., 2020).

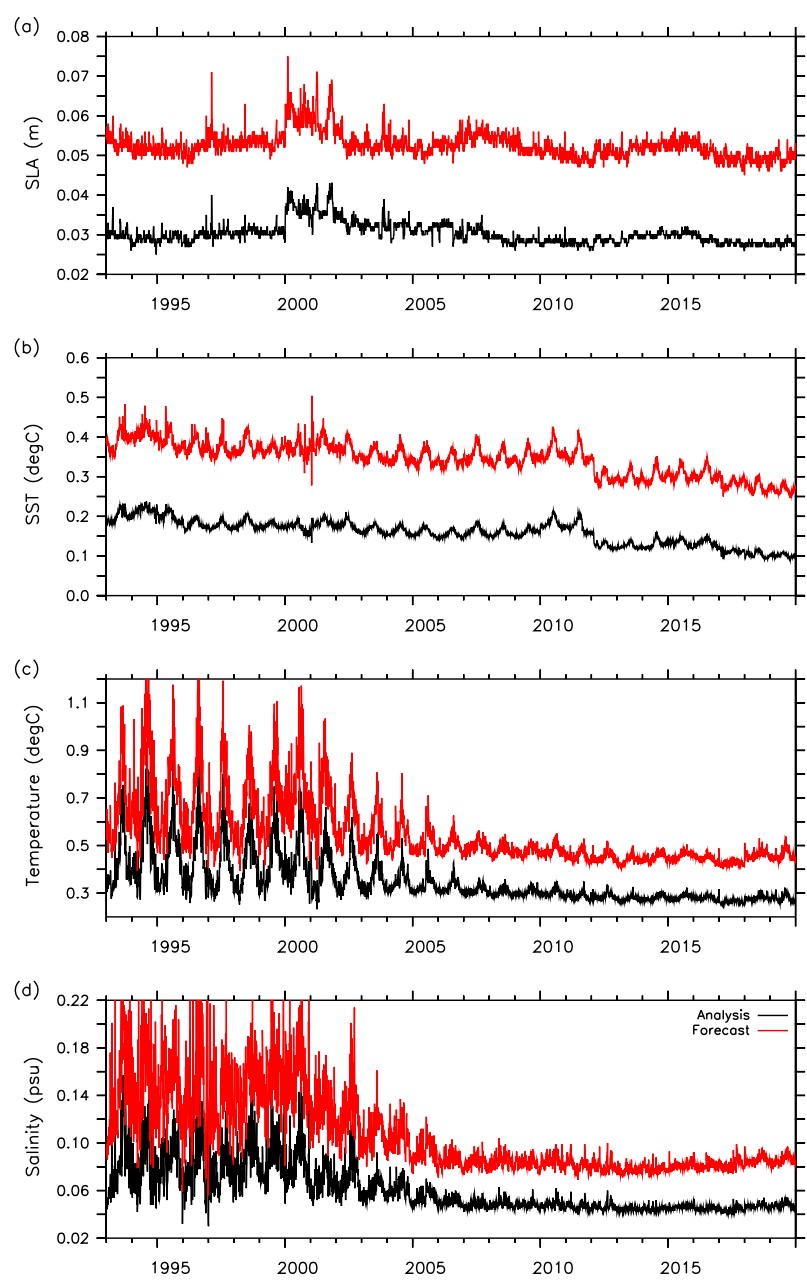

**Figure 2.** Time-series of global averaged mean absoute values of analysis (black) and background (red) innovations for (a) SLA, (b) SST, (c) *in situ* temperature, and (d) *in situ* salinity.

The analysis of innovations computed during the assimilation steps of BRAN2020, presented above, is intended to give an indication of the quality of the BRAN2020 product. A few caveats regarding these comparisons are warranted. A long, centered

observation window is used for sub-surface observations in the coarse-resolution data assimilation step of BRAN2020 (see
Appendix A). This means that some of the observations included to calculate the background innovations are assimilated in previous assimilation cycles. For SST and SLA, the observations used for the background innovations are all independent - having not been used to constrain the reanalysis. But for sub-surface temperature and salinity, this is not always the case.

Regarding the comparisons between BRAN2020 and BRAN2016, we again note that the data sources and volume of data assimilated in each experiment are different, and the reanalyses are initialised and forced differently. To more clearly attribute
the improvements in BRAN2020, Chamberlain et al. (2021a) tested a multiscale data assimilation system using precisely the same observations and boundary conditions as BRAN2016. Their results show substantial improvements in sub-surface temperature and salinity with multiscale assimilation relative to the BRAN2016 configuration, and smaller changes in surface metrics of the ocean state (SST and SLA). We therefore conclude that the main improvements evident in BRAN2020, compared to BRAN2016, relate to the new multiscale data assimilation approach presented in Chamberlain et al. (2021a).

**3.2  Assessment of daily-averaged reanalysed fields**

*Point-wise comparisons*

The analysis of innovations presented above are based on metrics calculated during the data assimilation steps of BRAN2020. However, the BRAN2020 dataset that is publicly available is comprised of daily-averaged, reanalysed fields that are produced by the model in between analysis updates. To assess these fields explicitly, we now compare the daily-averaged fields with
observations for each day of BRAN2020; 9861 daily comparisons from the start of 1993, to the end of 2019.

For SLA and SST, and for sub-surface temperature and salinity in depth ranges of 0-50m and 50-500 m, we group the model-data differences into 10x10° bins for the period 2008-2019 across the entire model domain. The results for each of these fields are presented in Figs. 3, 4, 5, and 6. For each figure, the top panel shows the map of Mean Absolute Deviation (MAD) between observations and daily-mean reanalysed fields; and the bottom panels show the difference between the MAD on day 3
(the day before each assimilation step) and day 1 (the day after each assimilation step) of each analysis cycle. These fields are intended to demonstrate the level of agreement between BRAN2020 fields and observations, and how differences grow in each analysis cycle. We expect that this should provide users with a clear idea of how accurate BRAN2020 fields are for all regions.

The MADs between reanalysed and observed SLA on day 1 of each assimilation cycle average 4.5 cm globally, with values of 6-12 cm in western boundary currents (WBCs), 4-5 cm along the Antarctic Circumpolar Current (ACC), and 2-3 cm in
equatorial regions and in the more quiescent parts of the ocean (Fig. 3). These MADs tend to grow by 1-2 cm in WBCs, by less than 2 cm along the ACC, and by less than 1 cm in the equatorial and less variable regions of the ocean.

For SST, The MADs between reanalysed and observed fields on day 1 of each assimilation cycle average 0.26°C globally, 0.2-0.6°C in WBCs, 0.3-0.4°C along the ACC, and with MADs of less than 0.2°C in equatorial and quiescent parts of the ocean (Fig. 4). These MADs tend to grow by 0.1-0.3°C in WBCs and along the ACC, and by less than 0.05°C for much of
the equatorial ocean. Unlike the growth in SLA, MADs for SST grow significantly in eastern boundary current regions, with increases of 0.1-0.2°C.

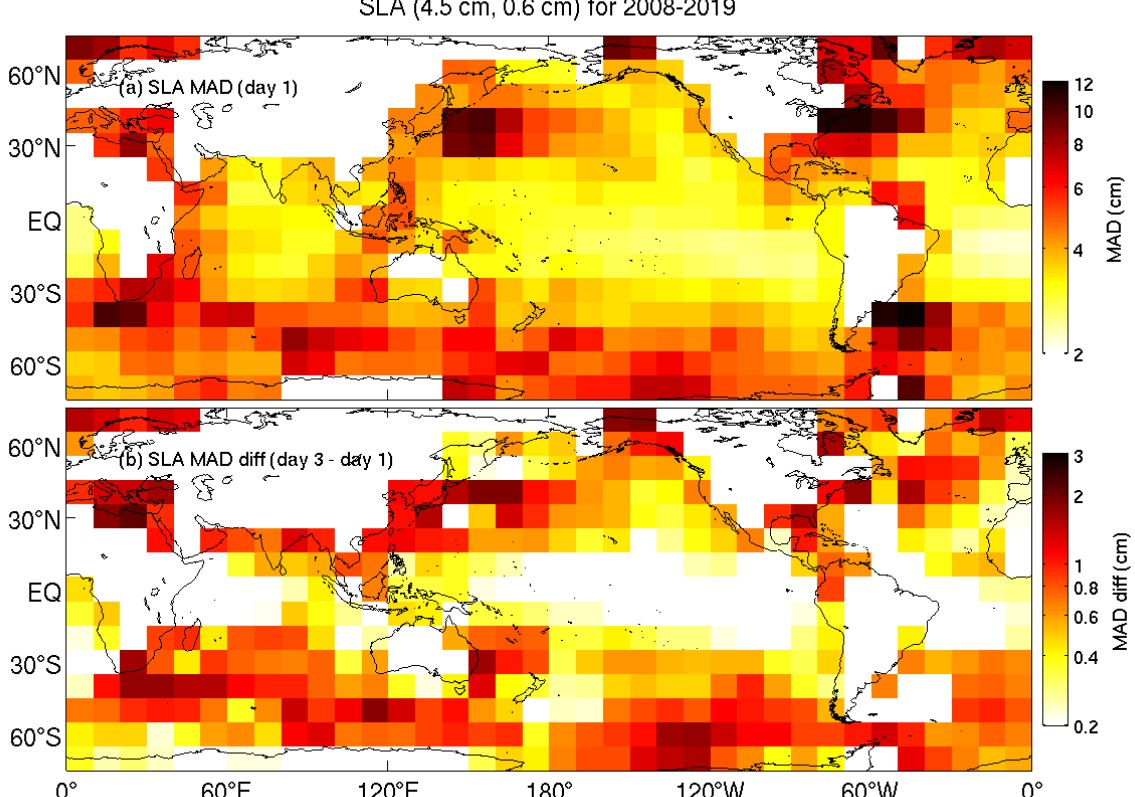

**Figure 3.** Binned sea level mean absolute deviations, averaged over 2008-2019 from the first day (top) and day 3 minus day 1 (bottom).

Figures 5 and 6 show similar distributions of MAD and MAD growth within the assimilation cycle for temperature and salinity, in two depth bands, 0-50 m and 50-500 m. For temperature in the top 50 m, the MADs between reanalysed and observed fields on day 1 of each assimilation cycle average 0.31°C globally, with values of 0.4-0.8°C in WBCs, values of

less than 0.3°C along most of the ACC, and less than 0.2°C in the equatorial and quiescent ocean (Fig. 5). These MADs for temperature in the top 50 m tend to grow by 0.2-0.3°C in WBCs, by 0.1-0.2°C along the ACC, and by less than 0.1°C for much of the equatorial ocean. Like SST, the upper ocean temperature MAD growth by day 3 is larger in eastern boundary currents, with increases of up to 0.3°C.

For temperature between 50 and 500 m depth, the MADs between reanalysed and observed fields on day 1 of each assimi-

lation cycle average 0.4°C globally, with values of 0.4-0.6°C in WBCs, less than 0.3°C along most of the ACC, about 0.5°C in equatorial regions, and 0.3°C the quiescent ocean (Fig. 5). These MADs tend to grow by 0.3-0.6°C in WBCs, by 0.3-0.4°C along the ACC, and by 0.2-0.3°C for much of the equatorial ocean.

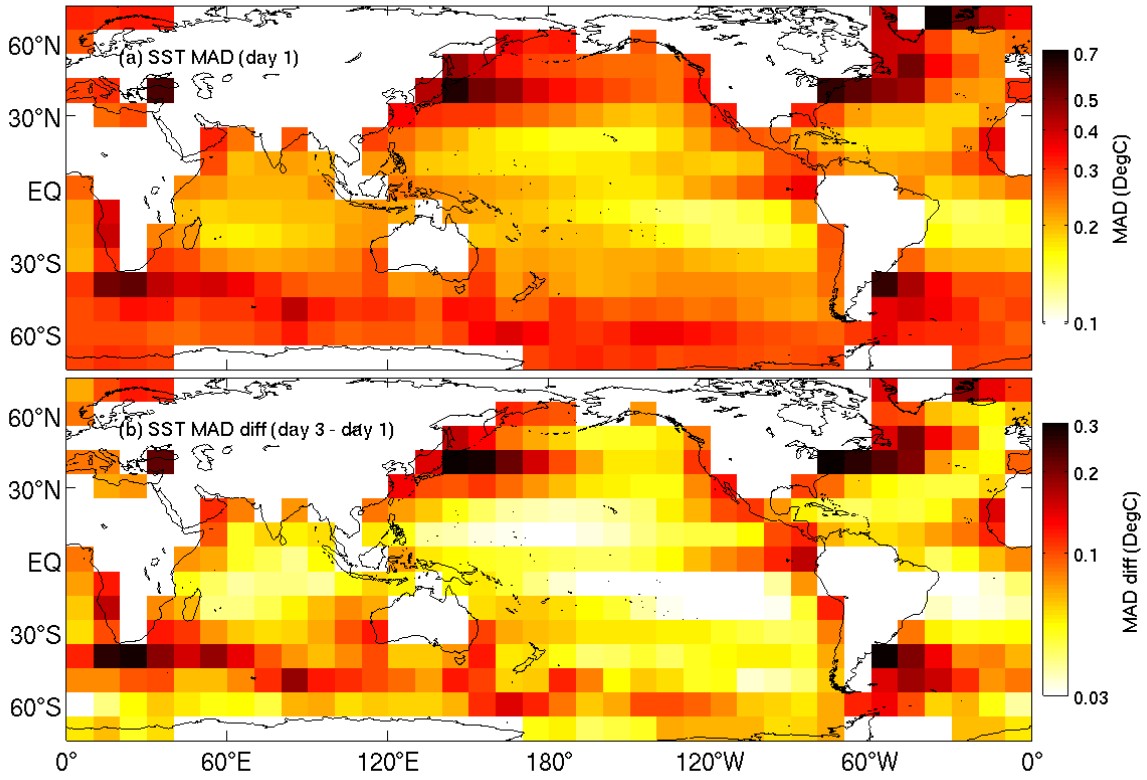

**Figure 4.** Binned SST mean absolute deviations, averaged over 2008-2019 from the first day (top) and day 3 minus day 1 (bottom).

For salinity in the top 50 m, the MADs between reanalysed and observed fields on day 1 of each assimilation cycle average 0.08 psu globally, with values of 0.04-0.2 psu in WBCs, less than 0.05 psu along most of the ACC, and less than 0.06 psu in equatorial and quiescent ocean (Fig. 6). These MADs for salinity in the top 50 m tend to grow by about 0.1 psu equatorward of 30°S and 30°N, and by less than 0.03 psu in the Southern Ocean.

For salinity between 50 and 500 m depth, the MADs between reanalysed and observed fields on day 1 of each assimilation cycle average 0.06 psu globally, with values of 0.04-0.2°C WBCs, values of less than 0.05 psu along most of the ACC, and values of about 0.06 psu in equatorial regions (Fig. 6). Those MADs for salinity between 50 and 500 m tend to grow by about 0.02-0.05 psu in the Pacific Ocean basin, by 0.02-0.1 psu in the Atlantic, by 0.02-0.07 psu in the Indian Ocean, and by less than 0.04 psu in the Southern Ocean.

Figure 7 summarises comparisons between reanalysed and observed temperature as a function of depth and time. Figure 7a shows profiles of sub-surface temperature MAD with depth for different time periods. Clearly shown is the decrease in MAD

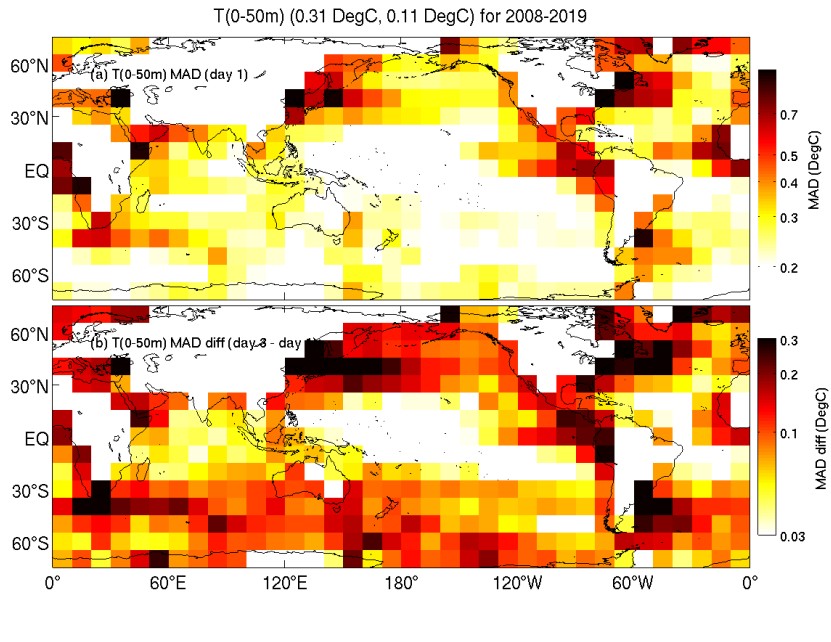

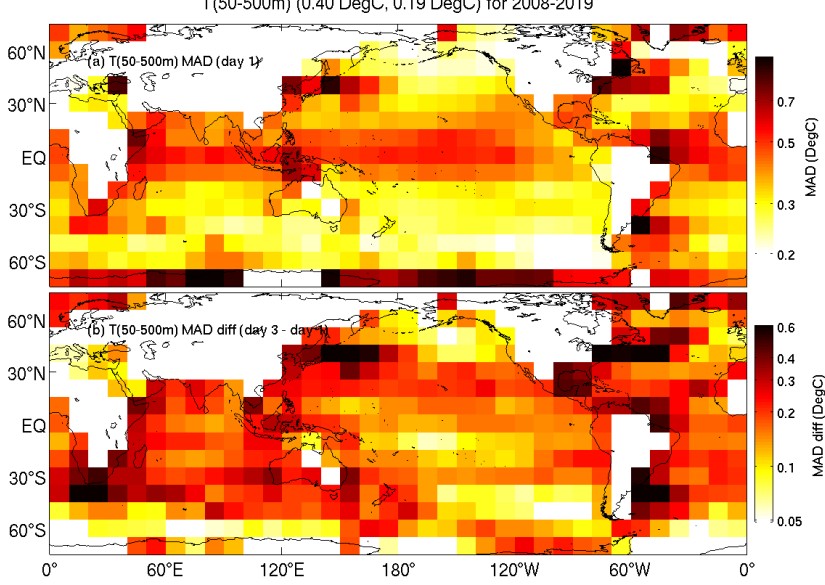

**Figure 5.** Binned mean absolute deviations of temperatures between 0 and 50 m (top pair) and 50 and 500 m (bottom), averaged over 2008-2019 from the first day (top of pair) and day 3 minus day 1 (bottom of pair).

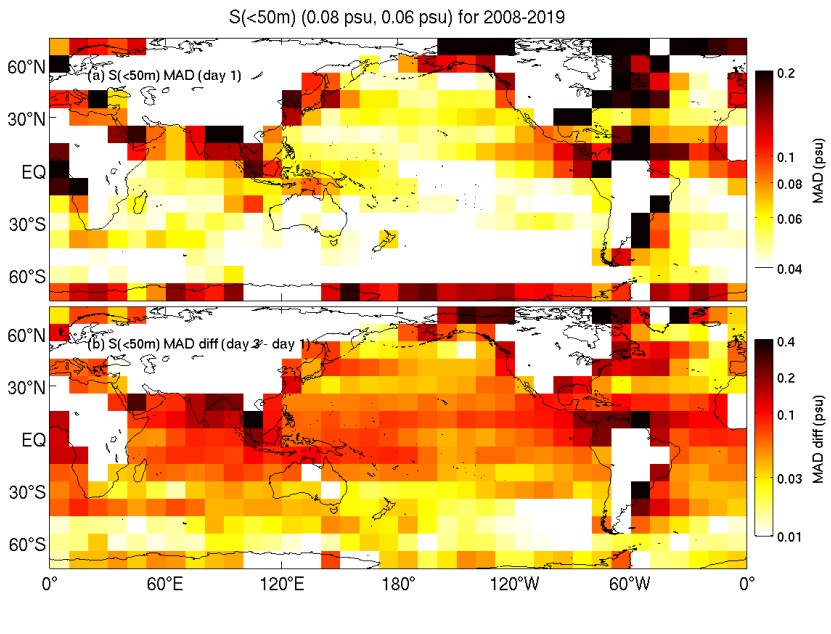

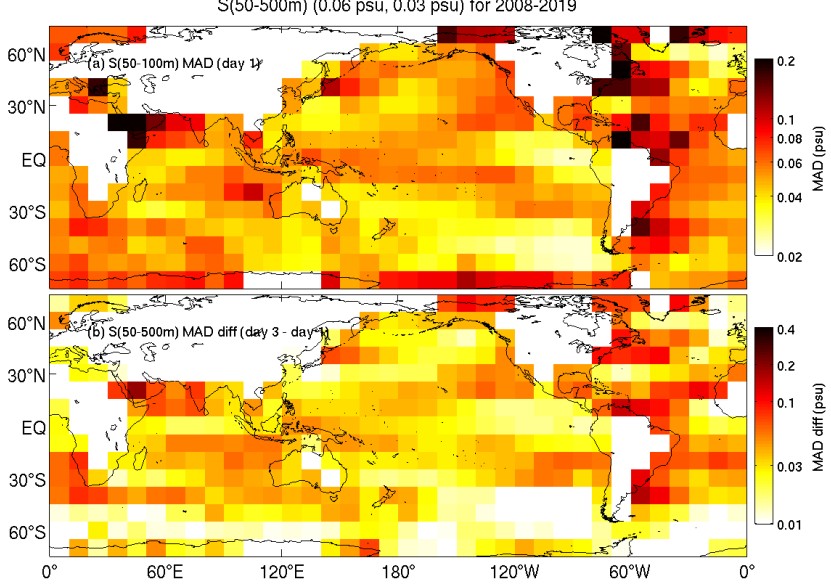

**Figure 6.** Binned mean absolute deviations of salinity between 0 and 50 m (top pair) and 50 and 500 m (bottom), averaged over 2008-2019 from the first day (top) and day 3 minus day 1 (bottom).

from the 1990s to the 2010s (Fig. 7a; compare the solid and dashed lines) as the number of sub-surface observations increase

with data from the Argo float array becoming available (Fig. 7c). Separate profiles of averaged MAD (Fig. 7a) are presented for regions of high- and low-variability, as indicated in the mask map (Fig. 7b; blue denotes high-variability regions), and averaged over times with and without Argo data. For this analysis, the regions of high variability are where the standard deviation of detrended SLA exceeds 8 cm. The choice of this cutoff value is arbitrary, but this value means that the high-variability region includes all of the WBC regions, the path of the ACC, and some of the high-variability tropical current systems.

Figure 7 shows that MAD is lowest in regions of low variability when Argo is available, and highest in high-variability regions pre-Argo. Figure 7d-f shows Hovmöller plots of model-data MAD fields for temperature for all regions (panel d), high-variability regions (panel e), and low-variability regions (panel f). For temperature, MAD is highest in the upper thermocline and below the surface ($\sim 50$-200 m) and a seasonal cycle is evident in the pre-Argo period, peaking in the middle of each year. MAD at the surface is relatively low, where reanalysed fields are well-constrained by SST observations. At depth ($> 500$ m)

temperature MAD are low everywhere, even at the beginning of BRAN2020, but reduce further with Argo data.

      The profiles of salinity MAD with depth (Fig. 8) are similar to those of temperature, with variability decreasing everywhere with Argo and increasing number of global observations. Without the equivalent of high-quality SST observations for salinity, the MAD is high at the surface. There is less difference in salinity MAD between high- and low-variability regions, compared to temperature; most of this difference is in the top 200 m.

*Sea-level variability*

      To further assess the variability of the reanalysed circulation, we compare estimates of sea-level variability, here quantified using the standard deviation of detrended sea-level (Fig. 9), for BRAN2020, a free-running model (Spinup-EI), and an observation-based analysis product (using Aviso-Ssalto/Duacs). Spinup-EI is a simulation with the same global ocean model used for BRAN2020, but is run without data assimilation (Oke et al., 2013a). The run used here is equivalent to the run that

is used to construct the high-resolution ensemble described in Section 2.3. The Aviso-Ssalto/Duacs product is a multi-mission product, generated by analysing along-track SLA observations to produce weekly maps of SLA on a 1/4° grid, and spans 1993 to 2014 (accessed from www.aviso.altimetry.fr); the sea level variability fields from models are calculated over the same years. The comparisons in Fig. 9 show that the simulated SLA variability is very similar for all products. The model simulation (Spinup-EI) realistically reproduces the key features of the observed field. However, we consider the comparisons between

BRAN2020 and the Aviso-Ssalto/Duacs product to be significantly better. The locations, spatial-extent, and amplitude of the local maxima of SLA variability in Fig. 9, agree more closely between BRAN2020 and Aviso-Ssalto/Duacs, than Spinup-EI. Figure 9 confirms that the reanalysed circulation in BRAN2020 is realistic and closely aligned with other observational estimates.

  *SST variability*

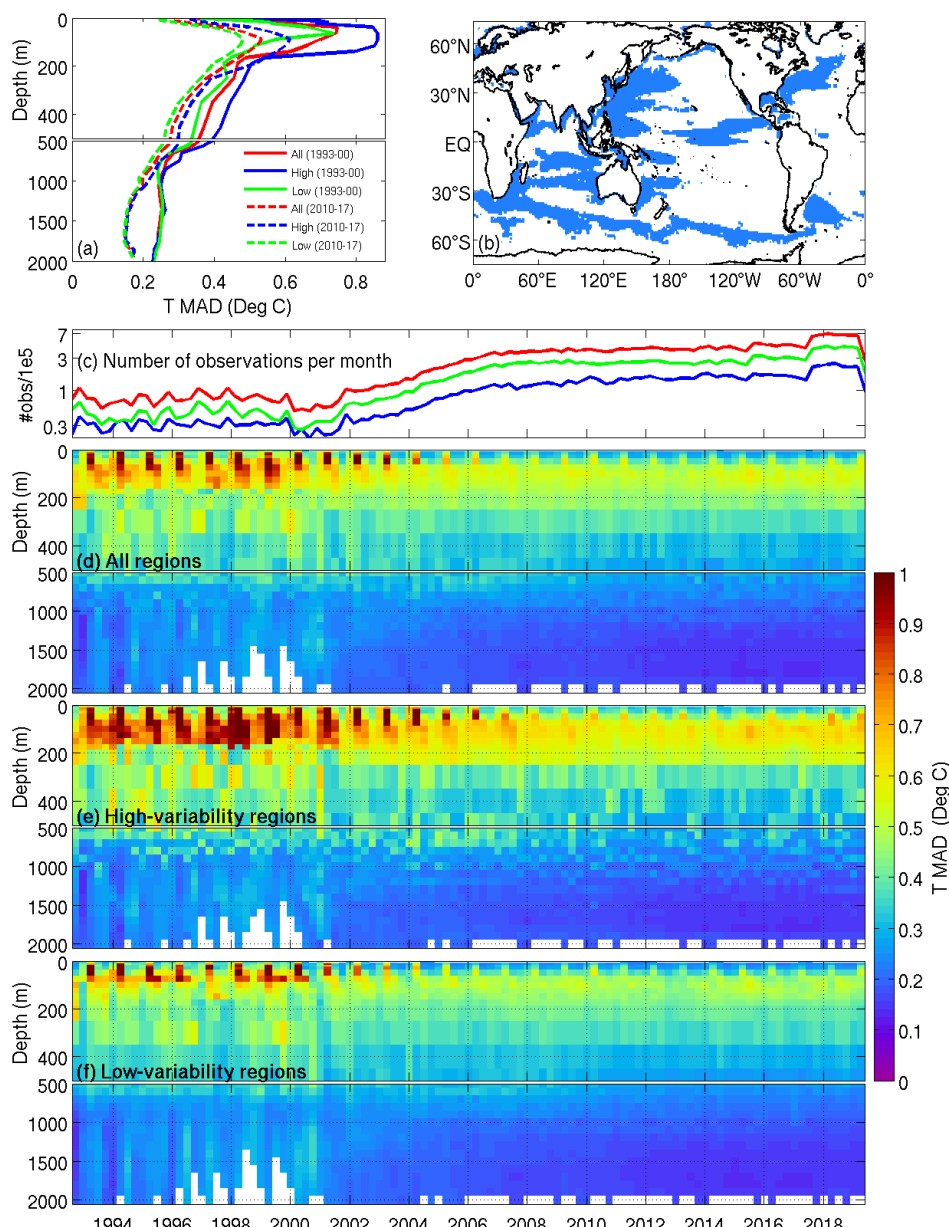

**Figure 7.** Mean absolute deviations (MAD) of temperature with depth over the course of BRAN2020. Results are shown for regions with high and low variability, as seen in sea level (shown top right), and combined. MAD profiles in the top left, averaged over times before the full Argo array and after. Time series of the number of observations over BRAN2020 (log scale) is above Hovmöllers of MAD for regions of low and high variability, and combined.

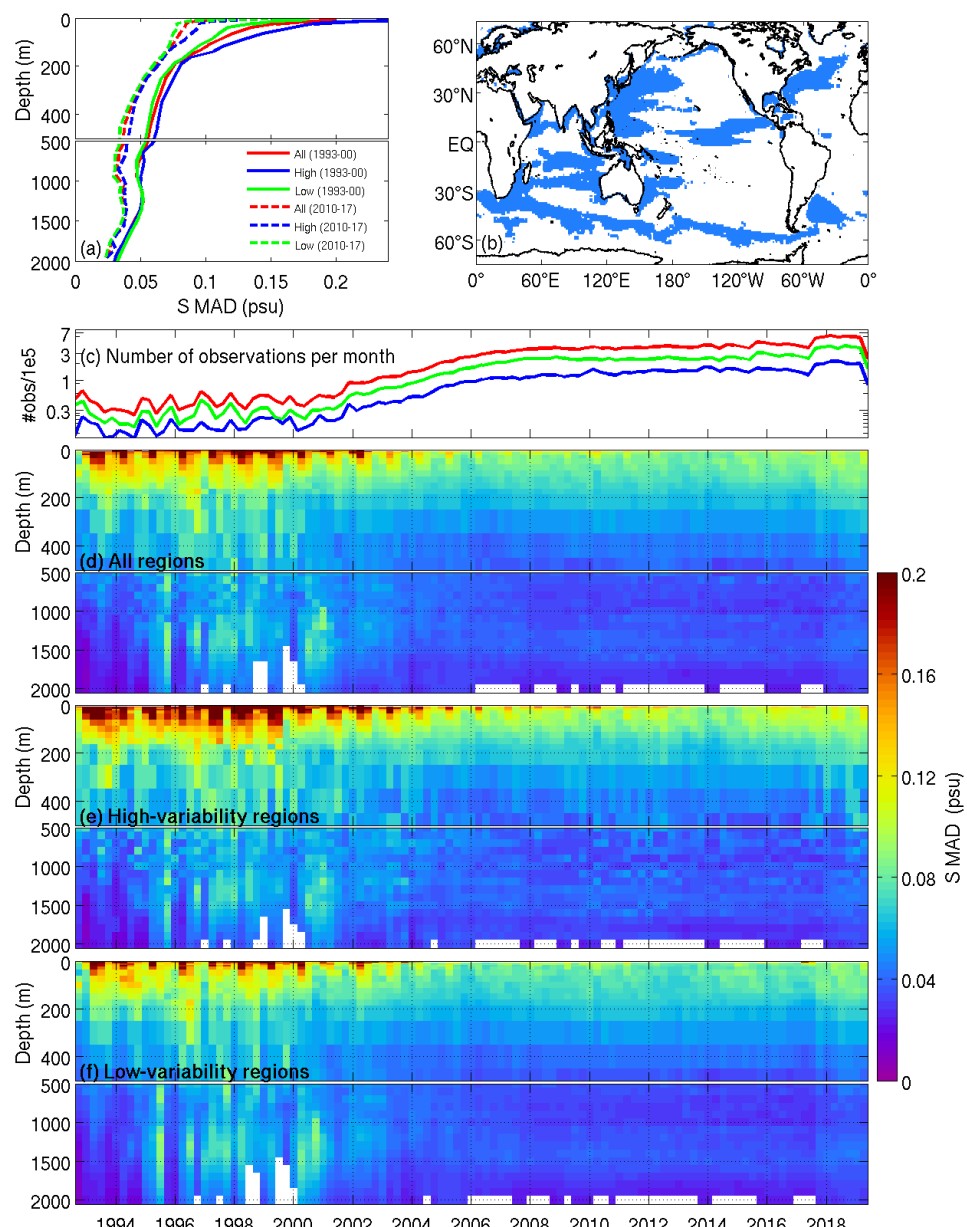

**Figure 8.** Mean absolute deviations (MAD) of salinity with depth over the course of BRAN2020. Same layout as Fig. 7.

Figure 10 shows a comparison of some key SST-based climate indices, calculated using observations (black) and BRAN2020 (red) fields. This includes comparisons of Niño3 and Niño4 using data from NOAA's Climate Prediction Center based on OISST (Banzon et al., 2014), as well as the Interdecadal Pacific Oscillation (IPO; as defined by Henley et al., 2015) and the

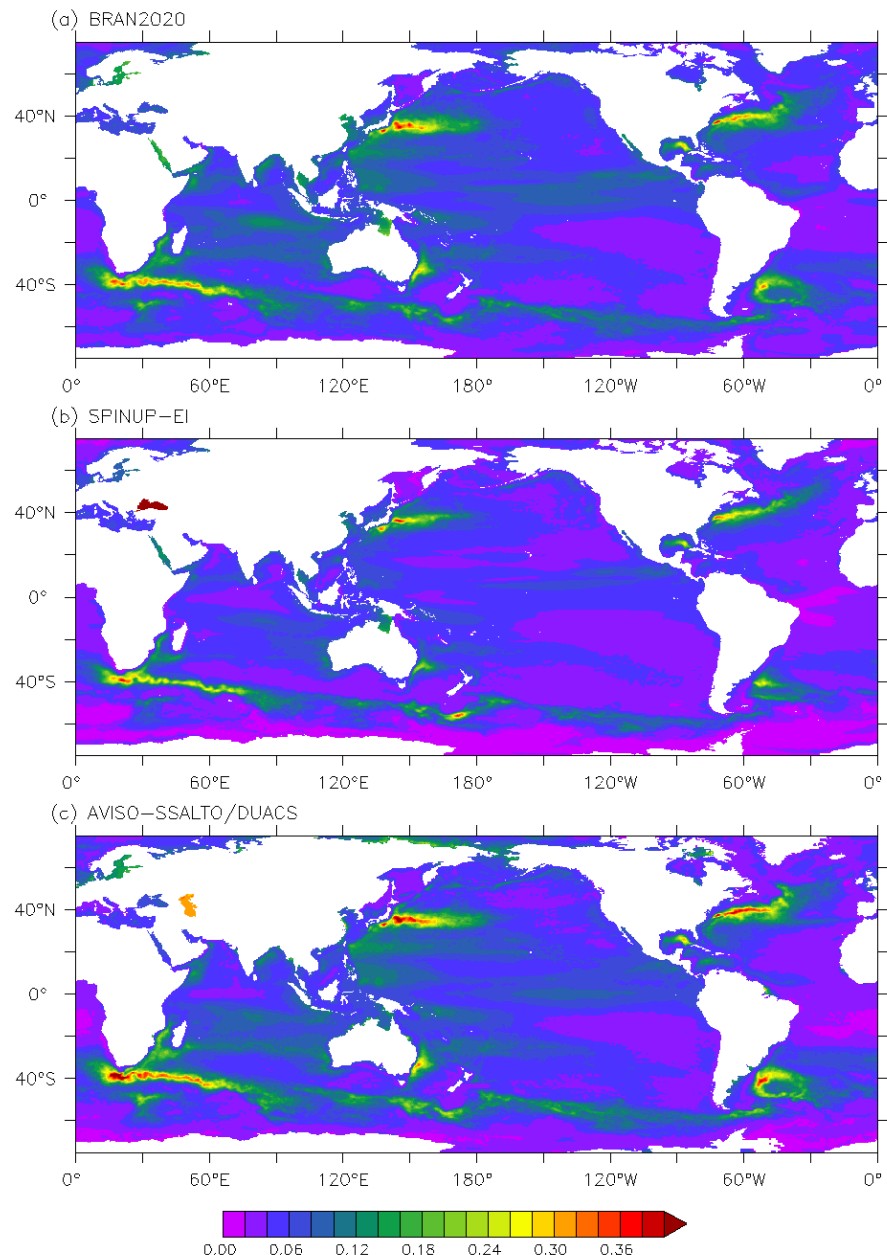

**Figure 9.** Sea level variability over 1993-2014 (detrended) from BRAN2020 (top), a model without data assimilation (middle) and observations (bottom).

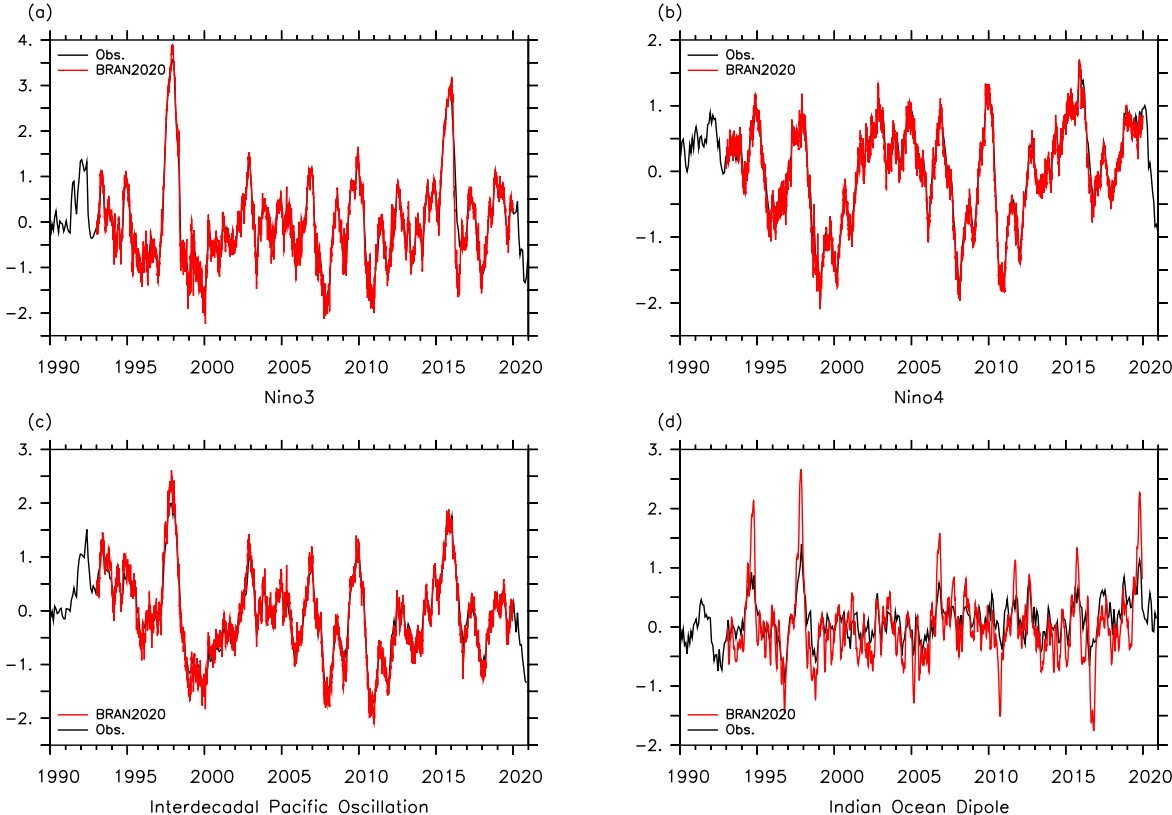

**Figure 10.** Major climate SST indices a) Niño3 (5°S-5°N, 210°E-270°E), b) Niño4 (5°S-5°N, 160°E-210°E), c) Interdecadal Pacific Oscillation (Henley et al., 2015), and d) the Indian Ocean Dipole (10°S-10°N, 50°E-70°E; 10°S-0°N, 90°E-110°E).

Indian Ocean Dipole (IOD) based on HadISST (Rayner et al., 2003). There is a very close match between the observed and reanalysed Niño indices and the IPO (Fig. 10a-c). The agreement for the IOD (Fig. 10d) is not as good. For this comparison, the observed IOD is calculated from a 1°-resolution HadISST product. The discrepancies shown in Fig. 10 for the IOD are mostly associated with the eastern node, a significant part of which crosses islands of Indonesia (see the boxes in Fig. 11a). SST estimates in different products depend on resolution, particularly in coastal regions with high variability. As a result, the details of the "observed" and reanalysed IOD differ in detail. Despite this, the observed and reanalysed IOD are still very consistent.

Figure 11 shows maps of SST variability, here quantified as the standard deviation of SST anomalies, for three products: BRAN2020, Spinup-EI, and gridded CCI-SST observation product (version 2.1, Merchant et al., 2019; Good et al., 2019). All estimates are calculated for the period spanning 2000-2009, with respect to climatologies from the same period. SST variability is greatest in regions of significant mesoscale eddy variability, such as WBCs and along the path of the ACC. SST variability is also high in places associated with interannual modes, such as the equatorial regions and the midlatitudes, which include the nodes of the SST indices indicated. Figure 11 shows that the SST variability simulated by a model without data assimilation

is realistic. But the comparison between the observation-based estimate (CCI-SST) and BRAN2020 shows exceptionally good agreement. In practice, it's difficult to identify any region of disagreement for the colour scales shown here. The only subtle differences, evident in this comparison, is in regions of seasonal sea ice cover, which BRAN does not include.

*Australian boundary currents*

Here we calculate volume transports along some of the major boundary current systems in the Australiasain region and compare results with previous versions of BRAN. Currents and velocities are not assimilated directly into the ocean state, making these comparisons a useful evaluation of the whole ocean reanalysis system. In this case, we compare transport from BRAN2020 with previous versions of the reanalysis and also with the "free-running" SPINUP-EI experiment without data assimilation, albeit with data assimilated into surface forcing driving the model.

Schiller et al. (2008b) examined boundary currents Australia from an earlier version of BRAN, presenting the average transports at several sections of the Indonesian Through Flow, the Leeuwin, the South Australian Current and the East Australian Current. The position of these sections are shown in Fig. 12 with long term average of currents in the upper ocean. Volume transport averages and ranges are shown again in Table 5, along with equivalent transports calculated from BRAN2020 both over the time corresponding to the Schiller et al. (2008b) analysis and the full time span available in BRAN2020. Since the reanalysis simluates mesoscale dynamics of the ocean, the variability and range of transports are large and new transport values from BRAN2020 fall well within the ranges of Schiller et al. (2008b) and in most cases the difference in transports from the two models are significantly less than the range.

Figure 13 shows time series of volume transports through a few selected selctions, comparing BRAN2020 transports with the recent BRAN2016 and a free-running experiment with the same ocean model setup. Significant multi-year variability is evident in the volume transport through Timor Strait. While the free-running model captures the seasonal cycle and even much of the year-to-year varability (due to climate signal used surface forcing), only the reanalyses with data assimilation capture the decreasing trend in westward flow over the years 2000 to 2015, which BRAN2020 shows to strengthen again in 2016. This decreasing trend is is also present in BRAN2016, albeit with a weaker magnitude.

The position of the transport shown for the EAC (bottom panel, Fig. 13) is past the point where it separates from the coast and part of the EAC extension, where the volume transport is less than the flow north of the separation point. While the mean of the 3 experiments shown are consistent, there is more variability in the BRANs relative to the free-running model, indicating eddies and mesoscale variability are not as prominent at this position in the ocean model without data assimilation. Also, southward transport here is stronger than mean reported by Schiller et al. (2008b), Table 5.

Interannual variability in the Leeuwin current system is less than variability from short time scales here, and the mean transport in the Leeuwin is weaker than the other systems presented here. The transport from the reanalyses show larger extremes relative to the free-running experiment. There are some persistent extremes in the BRAN transports, such strengthening southward flow in years 1999 and 2008; neither of which are captured in free-running model.

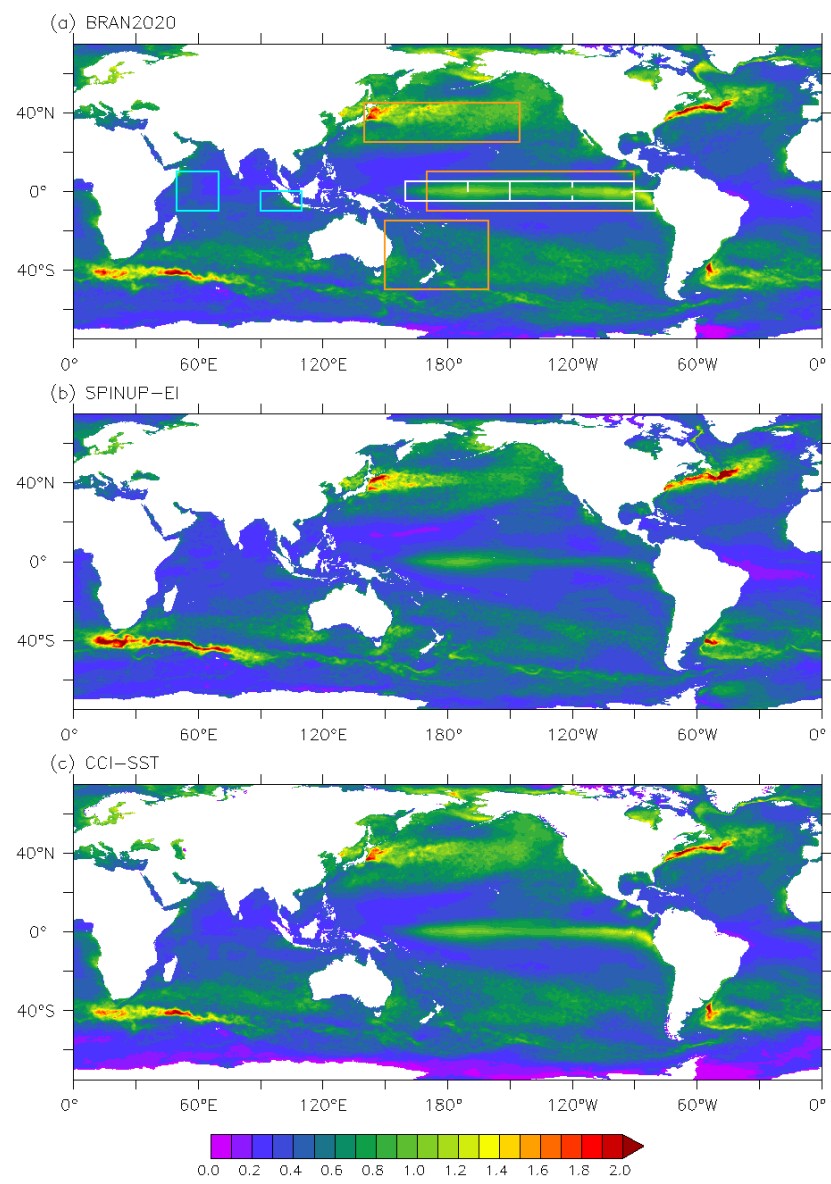

**Figure 11.** SST variability as standard deviation of climatological anomalies in BRAN2020 (top), Spinup-EI without data assimilation (middle) and CCI-SST (bottom) over the years 2000-2009. Top panel shows location for climate indices shown in Fig. 10; ENSO indices in white, IPO in orange and IOD in blue.

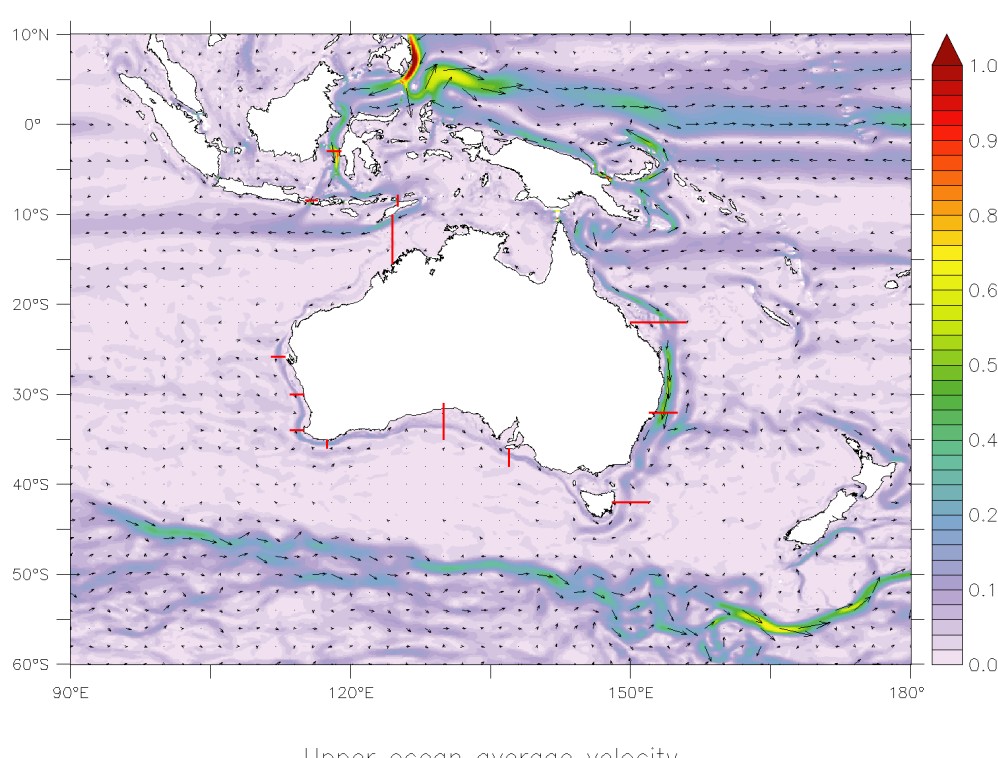

**Figure 12.** Average currents around Australia, averaged over 0-250 m, from years 2000 to 2009, and red lines indicate locations of transects listed in Table 5.

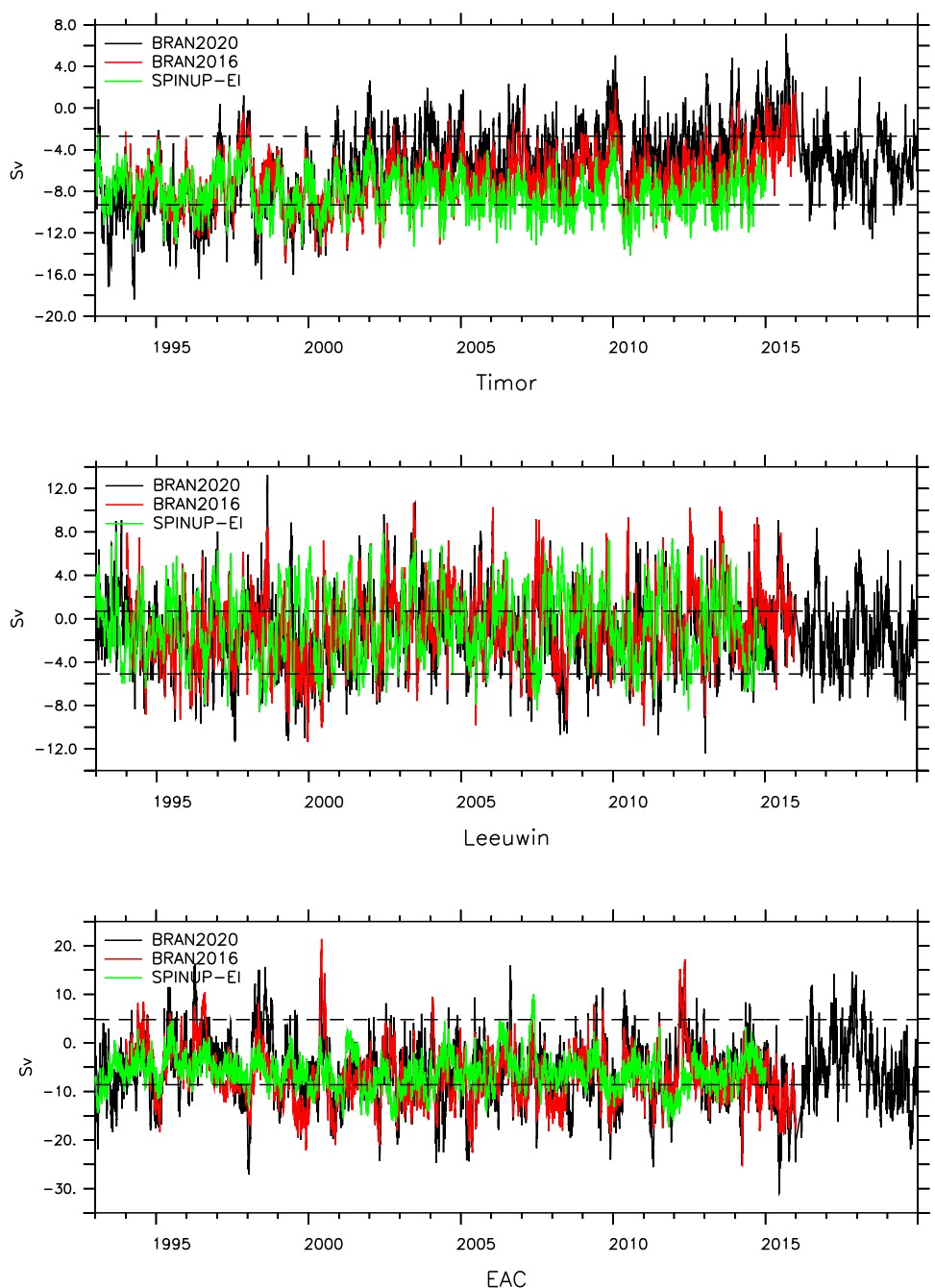

**Figure 13.** Time series of volume transport in Timor Strait (top), the Leeuwin Current at 30°S (middle) and the East Australian Current at 42°S (bottom) for BRAN2020, BRAN2016 and the free-running model (Spinup-EI).

**Table 5.** Transport, in Sv ($10^6 \mathrm{m}^3 \mathrm{s}^{-1}$), of various boundary current systems in the Australian region with results from Schiller et al. (2008b). Positive (negative) transport are northward or eastward (southward or westward). Uncertainties are 1 standard deviation of daily transport values. Transect positions are indicated in Fig. 12.

| Position | Schiller et al. (2008a) | BRAN2020 1993-2006 | BRAN2020 1993-2019 |
|---|---|---|---|
| ITF (full section integrals) | | | |
| - Makassar, 3°S | -8.4 ± 3.8 | -8.5 ± 2.3 | -7.9 ± 2.4 |
| - Lombok, 8.4°S | -1.5 ± 1.2 | -1.7 ± 1.4 | -1.7 ± 1.4 |
| - Ombai, 125.1°E | -1.5 ± 2.1 | -1.8 ± 2.1 | -0.9 ± 2.3 |
| - Timor, 124.5°E | -6.0 ± 3.3 | -7.0 ± 3.6 | -5.6 ± 3.6 |
| Leeuwin (0-300m) | | | |
| - 25.8°S, 111.5-113°E | -1.8 ± 2.3 | -1.9 ± 2.8 | -1.8 ± 2.9 |
| - 30.0°S, 113.5-115°E | -2.2 ± 2.9 | -1.2 ± 3.4 | -1.3 ± 3.3 |
| - 34.0°S, 113.5-115°E | -3.1 ± 2.6 | -2.3 ± 3.1 | -2.3 ± 3.2 |
| SAC | | | |
| - 117.5°E, 35-36°S, 0-300m | +1.2 ± 2.7 | +1.6 ± 2.8 | +1.5 ± 2.6 |
| - 130.0°E, 31.5-35°S, 0-1200m | -0.8 ± 4.9 | -1.4 ± 3.4 | -1.2 ± 5.5 |
| - 137.0°E, 36-38°S, 0-300m | +0.8 ± 2.5 | +0.5 ± 2.9 | +0.4 ± 2.9 |
| EAC | | | |
| - 22°S, 150-156°, 0-600m | -8.2 ± 6.7 | -9.6 ± 6.9 | -9.7 ± 6.8 |
| - 32°S, 152-155°, 0-1000m | -28.7 ± 23.5 | -27.3 ± 23.0 | -29.8 ± 22.8 |
| - 42°S, 148-152°, 0-1000m | -1.9 ± 6.7 | -5.6 ± 6.5 | -5.9 ± 6.6 |

## 4 Data availability

Output from BRAN2020 is available from NCI (Australia National Computing Infrastructure) OpenDAP servers at DOI:10.25914/6009627c7af03 (Chamberlain et al., 2021b). Daily averages of temperature, salinity, velocity, and mixed layer depth are available in monthly files in netCDF format. Three components of velocity are provided, however, the component of vertical was not saved in the first 5 years of BRAN2020. Monthly and annual averages of the same variables are also available.

Creation of the BRAN2020 dataset required significant resources. Apart from the human resources, BRAN2020 required up to 1200 cpus, 1 TB of memory, and executed over a period of about six months. The BRAN2020 dataset itself occupies ∼15 TB of disk-space.

# 5 Conclusions

For most metrics, BRAN2020 outperforms all previously presented reanalyses in this series of BRAN products. We attribute the improved performance of BRAN2020 mainly to the adoption of multiscale data assimilation (Chamberlain et al., 2021a). This delivered the greatest benefit for sub-surface temperature and salinity, where there are previously-acknowledged biases (Oke et al., 2013b). As a result, agreement between observed and reanalysed temperature and salinity fields below 50 m are of the order of 30% less in BRAN2020, compared to previous versions of BRAN. The learnings from performing BRAN2020 and the multiscale data assimilation paper (Chamberlain et al., 2021a) were applied to develop a new observational product, called Blue Maps (Oke et al., 2021). In that paper, Oke et al. (2021) demonstrate again the benefits of using EnOI with a diverse ensemble of anomalies that represent multiple time- and length-scales.

In the past, data from BRAN experiments have been used for studies of ocean general circulation (e.g., Schiller et al., 2008b; Chiswell et al., 2015; Feng et al., 2016; Oke et al., 2019); regional oceanography (e.g., Schiller et al., 2009; Oke and Griffin, 2011; Pilo et al., 2015; Schiller and Oke, 2015; Oke et al., 2018); the impacts of different components of the ocean observing system on data-assimilating models (e.g., Oke and Schiller, 2007; Oke et al., 2009); and applications to other fields, such as fisheries (e.g., Hartog et al., 2011) and marine biodiversity (e.g., Coleman et al., 2011). We offer the BRAN2020 dataset for use in similar applications.

## Appendix A: Ensemble Optimal Interpolation Configuration

For BRAN2020, EnOI is applied in a two-step multiscale data assimilation approach, as described in Section 2.1 and by Chamberlain et al. (2021a). Here, we present the technical details of the data assimilation, with specific reference to the parameters and settings (in Table A1) used with the open-source EnKF-C software (Sakov, 2014).

The details in Table A1 include the data assimilation cycle length, the ensemble size for each assimilation step, the method for ensemble construction, the localisation radii, the stride, the R-factor, K-factor, observation windows, and the domain over-which observations are assimilated. Each of these points are explained below. The data assimilation parameters are different for the two data assimilation steps; values from the high-resolution step are unchanged from the previous version, BRAN2016. The values used for BRAN2020 are those found to give good performance in tests of the multiscale data assimilation system in Chamberlain et al. (2021a)

The cycle length for BRAN2020 is 3 days; meaning that the ocean state is integrated forward 3 days between each application of data assimilation. Data is assimilated at both resolutions in each cycle. The ensemble size for the coarse- and high-resolution data assimilation steps are 480 members and 144 members respectively. The ensemble for the coarse-resolution assimilation is intended to represent broad-scale anomalies. The ensemble for the high-resolution assimilation is intended to represent anomalies associated with mesoscale features. In practice, we would like to make the ensemble sizes as large as possible. These are limited by computational resources. Larger ensembles are feasible for the coarse-resolution data assimilation step, as the ocean state vectors and ensemble members are much smaller and relatively inexpensive computationally compared to the high-resolution data assimilation step. The use of two data assimilation steps increases the effective ensemble size but also

**Table A1.** Data assimilation parameters at each resolution of multiscale data assimilation system for BRAN2020. (* - note, in practice, the observations windows were 1 day longer from the start of BRAN2020 to 2007.)

| | Coarse data assimilation | High-res. data assimilation |
|---|---|---|
| Cycle length (days) | 3 | 3 |
| Ensemble size | 480 | 144 |
| Ensemble generation | monthly climatological anomaly | 3-day minus 3-month centered differences |
| Loc. radius (km) | 1600 | 250 |
| Stride | 1 | 3 |
| R-factor | 4 | 1 |
| K-factor | 1 | 2 |
| Updated domain (latitude) | 65°S -65°N | 75°S -75°N |
| Argo obs. window (days*) | 10 | 3 |
| Non-Argo obs. window (days*) | 3 | 3 |

increases the diversity in ensemble members used, from the different scales of anomalies captured in the different resolutions. Both these ensemble aspects, size and diversity, are shown to improve data assimilation performance in an observational product, Blue Maps (Oke et al., 2021).

For the coarse-resolution assimilation, data is only assimilated within 65°S and 65°N, to avoid impacts of sea ice that are represented in the coarse-resolution ensemble. For the coarse-resolution assimilation step, *in situ* data are assimilated for a 10-day data window, centred on the analysis time. For all other observations, data are assimilated for a 3-day centred window. The 10-day *in situ* data window matches the cycle of Argo floats, so that this step can assimilate data from the full Argo array when available. Note, the data window is greater than the reanalysis cycle so some data are assimilated in 3 consecutive cycles in the coarse data assimilation step. We see no evidence of overfitting (degradation in forecast/background innovations) from this. Rather, increasing the density of *in situ* observations is beneficial in constraining the subsurface ocean that is poorly sampled and subject to dynamics and bias.

Following the equations presented in Section 2.1, the correction term applied to the the ocean state, to reduce the misfit to observations, can be written as a sum of the ensemble members, or:

$$\boldsymbol{w}^{inc} \quad = \quad \sum_{i=1}^{n} \boldsymbol{c}_i(x,y) \mathbf{A}_i(x,y,z), \tag{A1}$$

The values of $\boldsymbol{c}$ are related to the relative uncertainties of the model and observations, with higher magnitudes of $\boldsymbol{c}$ for higher model uncertainty, or lower observation uncertainty. The magnitude for a particular member ($\boldsymbol{c}_i$) is higher when it correlates with the background innovations. For further details of the procedure used here, please refer to Sakov (2014) and Oke et al. (2013b).

In theory, without localisation, $c$ (in Eq. A1) does not vary in space. But for most practical applications, with a large state dimension, this works poorly because the ensemble is rank-deficient - lacking a sufficient number of degrees of freedom to appropriately "fit" the background innovations (e.g., Oke et al., 2007). For BRAN2020, and most other applications of EnOI, localisation is used to increase the effective rank of the ensemble, and to eliminate unrealistic, long-distance ensemble-based covariances. This results in spatially inhomogeneous weightings that are a function of horizontal location, i.e., $c(x, y)$.

Use of localisation also means that observations beyond the localising length-scale from a grid point have no impact on analyses at that grid point. This is exploited by EnKF-C for computational efficiency, by projecting background innovations onto the ensemble separately for individual horizontal grid points. This means that instead of a single - practically impossible - calculation, the assimilation problem is reduced to a large number of smaller calculations, with one calculation at individual horizontal grid points. The localising function used by EnKF-C is a quasi-Gaussian function, where the function goes to exactly zero over the localisition radius. The effective $e$-folding length-scale of the localising function is $\sim3.5$ times smaller than the localisation length-scale reported here (and elsewhere in the literature). The localisation radius used with high-resolution data assimilation is 250 km, consistent with the scale of the mesoscale features that are corrected in this step. We found that increasing the localisation radius to 1600 km improved the efficacy of the course-resolution assimilation step by including more neighbouring *in situ* profiles in the assimilation at any point; noting that the typical spacing of Argo floats is $\sim300$ km.

It's expected that the weighting coefficients in $c$, vary smoothly in space, with values for adjacent grid points being very similar. This permits a further simplification with the use of "stride" (Sakov, 2014). In practice, weights are only calculated on a subset of grid points, determined by the stride; for instance, a stride of 3 for the high-resolution step means that coefficients are only calculated for every third grid point in each horizontal direction, significantly reducing the computational cost. When a stride greater than 1 is used, weights ($c$) are linearly interpolated onto the model grid before constructing the increments.

EnKF-C includes an option to use a so-called "K-factor," described in (Sakov and Sandery, 2017). The K-factor is intended to prevent overfitting, by modifying observation variances so that the corresponding increments do not exceed the specified value times the spread of the ensemble.

Another parameter used to tune the data assimilation performance of EnKF-C is the "R-factor". The R-factor scales the assumed observation error variances, affecting the ratio of the background error variance (the ensemble variance) to observation error variances, and adjusting the impact of observations on the calculated corrections to the ocean state (Sakov, 2014). The R-factor is typically increased for applications using larger localisation radii that include more observations for each calculation. This increases the relative assumed observation error variance, again, to avoid over-fitting. In practice, using an R-factor of 4, effectively reduces the amplitude of the ensemble variance by 4, which is the same as reducing the effective background error (ensemble spread) by $\sqrt{4}$.

*Author contributions.* MC and PO designed the experiments and MC carried them out. RF contributed to the development of the model. GB provided input to the design of the work. MC and PO prepared the manuscript with contributions from all co-authors.

*Competing interests.* The authors declare that they have no conflict of interest

*Acknowledgements.* Production of BRAN2020 is supported by the Bluelink Project, a partnership between CSIRO, the Australian Bureau of
Meteorology and the Australian Department of Defence. We thank Pavel Sakov for advice and assistance in data assimilation and EnKF-C.
Data was sourced from the Integrated Marine Observing System (IMOS, www.aodn.org.au). IMOS is a national collaborative research
infrastructure, supported by the Australian Government. CCI L2P SST data from AVHRR and ATSR sensors were produced at the Met
Office as part of the European Space Agency (ESA) SST Climate Change Initiative (CCI) Project, funded by ESA. AMSR-E L2P SST
data were provided by GHRSST and REMSS. AMSR2 L2P SST data were provided by GHRSST and JAXA/EORC. The AVHRR L2P
SST data from the Naval Oceanographic Office are made available under Multi-sensor Improved Sea Surface Temperature (MISST) project
sponsorship by the Office of Naval Research (ONR). The VIIRS L3U SST data were provided by Group for High Resolution Sea Surface
Temperature (GHRSST) and the National Oceanic and Atmospheric Administration (NOAA). Argo data were collected and made freely
available by the International Argo Program and the national programs that contribute to it. (www.argo.ucsd.edu, argo.jcommops.org). The
Argo Program is part of the Global Ocean Observing System. This research used computation resources and archives available at the National
Computational Infrastructure (NCI), which is located at the Australian National University and supported by the Australian Government.

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
