# Peer review of "Next generation of Bluelink ocean reanalysis with multiscale data assimilation: BRAN2020"

_Earth System Science Data, 2021_

## Author Comment (AC1)

**Comment on essd-2021-194**

Anonymous Referee #1

Referee comment on "Next generation of Bluelink ocean reanalysis with multiscale data assimilation: BRAN2020" by Matthew A. Chamberlain et al., Earth Syst. Sci. Data Discuss., https://doi.org/10.5194/essd-2021-194-RC1, 2021

The manuscript presents a detailed description and validation of an ocean reanalysis, and it is very appropriate to accompany the dataset itself. The paper is well written, and explains very honestly the strengths and weaknesses of such dataset. I enjoyed reading it. As such, I recommend the paper for publication, but I ask the authors to better discuss, illustrate and explain several issues, summarized below. I think having a dedicated section "Discussion" before the Conclusions is a good option.

Thank you for reading the manuscript and the useful comments. The manuscript is modified to address the issues raised, please see below for details.
Extra details have been added to the introduction which give further context to the work, and in a new subsection to the results.

**General points** (for which I suggest including a more detailed discussion).

- Neglecting sea-ice modeling in the reanalysis seems to me a weak point, considering the effects of sea-ice induced circulation around e.g. ACC, which is certainly an area of interest for the authors. A deeper discussion about what they expect to miss, and if there are plans for including a sea-ice model will benefit the paper.

A paragraph has been added to the introduction to give better context to the motivation and applications of the work presented.
The development of BRAN is in support of operational ocean forecasting around the Australian region, and as such, dynamics close to Antarctica influenced by sea ice processes have not been a priority.

Previous versions of BRAN have been the basis of many studies, some of which are now listed in the introduction as well. The Bluelink Project does intend to include sea ice in future versions of BRAN to increase the utility of the product for research.

- The validation mostly focus on observation diagnostics. In my opinion a reanalysis is unique in the sense that can capture integrated diagnostics (OHC trends, transports, etc.) which are to some extent unobserved. It seems from the presentation of the work that climate applications are not the focus at all of this dataset. Also, the heterogeneous assimilated data (many datasets switching from delayed to real-time mode) may also compromise the low-frequency variability. Any thoughts about this, to include in the Summary/Discussion?

Previous versions of BRAN have indeed found many applications, e.g. to study transports around the Australian region (Schiller et al 2008, and, Divakaran and Brassington, 2011), extreme temperature events (Schiller et al. 2009, and, Oke and Griffin 2011), as well as providing boundary conditions to regional models (Steven et al. 2019).
This is now mentioned in the introduction of the paper.
This new version of BRAN is also suitable to the same applications, with the advantage of reduced biases.
As a demonstration, a new subsection is added to the paper comparing boundary currents around Australia in the new reanalysis with previous estimates, and finds that the results are entirely consistent,

The impact of heterogeneous datasets is unavoidable in the production of long reanalyses.
Under the "Analysis of innovation" section and discussion of Fig. 2, there is commentary regarding the impact of new observations entering the reanalysis.
The most striking feature is the reduction of mean absolute innovations for subsurface temperature and salinity once coverage of Argo data becomes global.
Also noted is the improvement in SST as new satellites and sensors become available; with VIIRS in particular in 2012.

- The DA system is detailed in a companion paper, not available at the moment (in review for Ocean Modeling). Because of that, some aspects of the DA system are probably not explained, so the reader

may have troubles to understand the formulation, if there is some lag between the two publications (see also below specifically).

This paper describing the DA system is now accepted (Chamberlain et al. 2021, doi:10.1016/j.ocemod.2021.101849) and publicly available. Some of the key results are emphasised again here. In particular, the impact of multiscale data assimilation at separate scales, which is demonstrated in this companion DA paper with a clean set of experiments.

**Minor points**

Line 4: it is said up to 2019. It will be useful for the interested readership to specify the plans for update (near real time, once in a while, or never?).

Yes, the intention is to update BRAN2020 to within months of real-time while it is our most current configuration. Comments have been added to the abstract as suggested.

Line 7; "for some variables" please be specific in the abstract.

Text is clarified here to specify that it is the subsurface temperature and salinity as the variables that are most improved.

The spinup period seems short (3 months). Do you have evidence that is enough for stabilizing most low-frequency variability indexes, or was chosen more for practical reasons?

We find that after 3 months or 30 DA cycles there is no further improvement or decrease in the innovations calculated, indeed, most of the decrease is within the first month. This is now clarified in the text.
We also avoid a long spinup to reduce the amount of drift and bias to build up in the ocean state before data is assimilated.

Line 72 vertical resolution seems quite low compared to most other reanalyses. It is also not clear (Line 142) how can the system assimilate both night and day time SST data, since the diurnal cycle will be for sure underestimated. Or maybe I am missing something in the explanation (lines 141-142 are not very clear to me).

The diurnal cycle is essentially removed from both the background and the observations. The background from the model is a daily average, so even the dampened diurnal cycle in the model is removed. SST data density is high, and the process of calculating super-observations averages both day and night observations. In addition, as stated, the "sea surface temperature at 0.2m" is assimilated rather than "skin temperature," also reducing any diurnal cycle in the data.

Line 74: Forcing fields masked if sea-ice. Not clear how do you use them if sea-ice occurs? Which fluxes would you use instead?

A short explanation is added, "values in the atmospheric reanalysis fields are replaced with values expected below sea ice…"
This is applied in a preprocessing step before running the ocean model.

Line 108: it is clear from the text below that the ensemble anomalies in the EnOI are "climatological" (i.e. not flow-dependent). Better to specify here for clarity.

At line 108, it is clarified that there are two separate ensembles.
In the text below it stated the coarse ensemble contains climatological anomalies; the paragraph now also states the high-resolution ensemble is built with seasonal-scale anomalies.

The multiscale Data assimilation formulation seems sub-optimal: no proper scale separation is used, and the use of the same observational data between the two steps implies non-zero cross-covariances between observations and background (in the second step). This seems theoretically sub-optimal and should be mentioned. Another issue is that the time dimension does not change between the two steps. It would be reasonable to assume that longer (broad scale) dynamics is associated to longer time scales, while here the 3-day time scale applies to both. Any thought about this? Again, maybe this is included in the manuscript submitted to Ocean Model., but without it being published it is worth to discuss.

In the paper that has now been accepted by Ocean Modelling, the method is shown to effectively apply corrections at different scales, since the anomalies in each ensemble have different scales.
Features at all scales are present in the observations. The scales in the increments calculated at each step are determined when the DA

system projects the observation-model innovations onto ensemble members.
This is now discussed in the text at the end of section 2.3.2.

Broad scale dynamics might act on longer time scales. However, in practice, it was beneficial to apply the coarse DA correction at each analysis cycle due to drift and biases accumulating noticeably when the coarse DA step was not applied every cycle.

MDT (lines 155-157): I understand this is computed as Mean SSH from a free-running- model run. This means that long-term mean barotropic-dominated transports (e.g. in ACC) in the reanalysis should look very similar to the control experiment, by construction. Some comments about this will be beneficial, as most other reanalyses use other strategies (either an "observed MDT" or one with assimilation of in-situ data)

You are correct and comments are added to the text as suggested. This will be reconsidered in future versions.

Table 1. TEM vs TEM2 and SAL vs SAL2: the difference is not explained in the text.

Details added to text, "observation types with higher uncertainties, such as XBT and sea mammals, are assigned larger errors in types TEM2 and SAL2 in Table 1."

Line 180: Doesn't sound better/Isn't more common to say "super-obbing" with double "b"?

This short-hand term has been replaced with a fuller description, "to build super-observations."

Table 2,3,4. I really like the efforts in quantifying the error growth. Perhaps reporting all those values also in the figure 7,8 (in the profile panel) will help to better see the error growth and the differences with BRAN2016

The values in these Figures are calculated separately from those shown in the Tables. The error growth in Table 4 are differences in the magnitudes of forecast/background and analysis absolute errors from the DA step of the analysis cycle, whereas the profiles were calculated

separately based on observation-model differences from daily averaged ocean model output. These profile panels are already busy, and adding an error growth profile calculated from the DA cycle would also be inconsistent with the figure.
However, there is depth information still in the Tables with a breakdown of error growth near the surface (<50m), moderate depths (50-500m) and deeper (>500m). Values are consistent with the profiles, and are sufficient to compare and contrast error growth in subsurface temperatures and salinity.

Also, better to say immediately which observations are used to validate and form the statistics: are those assimilated in BRAN2016 only or also those supplemented in BRAN2020 (like sea mammals, etc.)? Both are possible choices in my opinion, but the interpretation of the results will be different.

The values in the tables are based on the observation databases assimilated into the respective versions.
This is now noted in the text in section 3.1.

Linked to this: lines 265-269 are not very clear to me. I don't understand why differences in skill scores improvements between surface and sub-surface data should lead to the authors' conclusions? It is because of the same atmospheric forcing/ingested surface data? Better to state clearly.

This last of paragraph of section 3.1 is rewritten to clarify.
We are attributing the improvement of the subsurface data assimilation to the multiscale technique based on results in Chamberlain et al. (2021a, doi:10.1016/j.ocemod.2021.101849), that is now published.
The comparison between BRAN2020 and BRAN2016 is complicated by other differences applied to the set up and observations.
However, in Chamberlain et al (2021a), a clean comparison was done where the only change was the addition of the second data assimilation step, and this showed similar improvements in the subsurface to those described in BRAN2020.

Line 323 typo in "reanalysed"

Fixed.

Line 331: SPINUP-EI is a misleading name. Perhaps change to CTRL or similar

Spinup-EI is one of the existing Bluelink spinup experiments that is also available from the NCI data catalogue, and has been used in other applications, so the nomenclature will be kept in this case.

---

## Author Comment (AC2)

**Comment on essd-2021-194**

Anonymous Referee #2

Referee comment on "Next generation of Bluelink ocean reanalysis with multiscale data assimilation: BRAN2020" by Matthew A. Chamberlain et al., Earth Syst. Sci. Data Discuss., https://doi.org/10.5194/essd-2021-194-RC2, 2021

Review of "Next generation of Bluelink ocean reanalysis with multiscale data assimilation: BRAN2020" manuscript.

The Next generation of Bluelink ocean reanalysis with multiscale data assimilation: BRAN2020" manuscript presents a new attempt to better estimate ocean dynamics in the multi-decadal global ocean arena. The structure of the manuscript is well organized and scientific ideas are correctly exposed.

Thank you for reading the manuscript and the useful comments.

General comment:

In the manuscript, the authors applied the EnKF-C method to propagate observations information inside the ocean model. In my opinion, omitting sea ice in the system is a major problem, it is an important phenomenon influencing dynamics in the Southern Ocean. Using analysis every 3 days seems too frequent for the global ocean model having a spatial resolution of 1/10 degree. For example, GLORY NEMO experiment is using a 14-days assimilation window. Some discussion along the lines would be necessary; how authors decided for 3-days as an appropriate assimilation window. Seems to me that it's quite short (or is quite often – every 3 days) time span between analysis, and is a way of imposing stiff control over the ocean system i.e. suppressing model physics to fully develop. Initialization of the temperature fields by using the daily averaged values seems strange, the model vertical resolution in the surface layers is 5m which is prohibiting diurnal oscillations to fully develop. In that sense, the model is not resolving diurnal SST dynamics (not sure about the temporal frequency of atmosphere forcing), and this shouldn't be the reason for using daily fields (which are dynamically unbalanced). As

assimilation is done on the two different scales, could the time step for applying analysis be different (longer for large scale and shorter for mesoscale)?

A paragraph has been added to the introduction to give a better context to the work done.
The development of BRAN is in support of operational ocean forecasting around Australia and has found many other applications across this broad region, now listed in the introduction as well. As such, BRAN does not focus on dynamics close to Antarctica or processes associated with sea ice at this stage.
However, the Bluelink Project intends to include sea ice in future versions of BRAN.

GLORYS12 has recently been published (Lellouche et al. 2021., Frontiers in Earth Science, doi:10.3389/feart.2021.698876) and is a similar configuration to BRAN, in domain and resolution. GLORYS12 used a data cycle of 7 days and overall obs.-model differences are very comparable, even though the two systems implement corrections differently; e.g., see panels (d) from Fig.s 7 and 8 here alongside panels C and D of Lellouche et al. 2021.
3-day cycles have been used in BRAN for several years (e.g. Oke et al. 2018, doi:10.1016/j.dsr2.2018.09.012), and now cited in Section 2.1.

In 3$^{rd}$ version of BRAN (Oke et al. 2013, doi:10.1016/j.ocemod.2013.03.008), the analysis cycle was reduced from 7 to 4 days, which was shown to substantially reduce both misfits to observations and magnitude of corrections applied.

The objective in these reanalyses is to follow the observed mesoscale dynamics as much as possible and not allow the model physics to drift too far.
Over this time frame, error growth is about linear, the longer the analysis cycle the further any model will drift from the observed ocean. The benefits of a shorter cycle lengths are smaller errors, at the cost of extra computation. Other free-running experiments might be better suited to studying internal ocean processes where model physics are free to develop.

To clarify, daily averaged temperatures are used just as background fields to the data assimilation (described in section 2.1). The

correction/increment calculated is applied back to the original, instantaneous restart, so that the model does simulate subdiurnal processes (even though they are not saved).  The temporal resolution of the JRA55-do forcing is subdiurnal.

During development of the multiscale DA system, applying coarse DA on a longer timestep was tested but it was found that errors accumulated due to model bias over the longer cycle.

Specific comments:

Line 54: Are authors referring to common term residuals of data assimilation when they talk about the difference between the analysis and observation?

The analysis innovations are similar to the 'residuals' as they are described in some papers.
'Also, referred to as "residuals,"'  is added to the text here.

Line 180: Not sure if this is a typo mistake: " super-obing". It sounds a bit strange, usually, we refer to "super-obs" or "super-observations".

This short-hand term has been replaced with the full term, "super-observations," as suggested.

Line 197: Analysis innovations are sensitive to the observation errors, and in that sense are the observation errors constant in space/time or they are varying (for specific observation type)? If not do authors think it would improve the assimilation system?

Different observation types are used to manage varying observation errors, these errors are constant within a type.  Note that there are different SST types in Table 1 with different errors that are assimilated into the reanalysis for years they were available.
We have captured most of the evolution of observation errors over the course of this new reanalysis, with the exception of AVHRR-SST which should have a larger error, as has been discussed in the manuscript. Also, as noted in the text, the observation error assigned here should include representation error as well as instrumental error, i.e., the uncertainty of a 'point' observation to represent the grid cell it is applied to (which is, as stated, poorly known).

---

## Author Comment (AC3)

**Comment on essd-2021-194**

Anonymous Referee #3

The manuscript is well written and presents relevant results and discussions. Thank you for dedicating time to publish a detailed discussion and analysis of this dataset. However, a more detailed discussion on a few topics would greatly benefit the manuscript.

Thank you for reading the paper and the comments.
The manuscript is modified to address the issues raised, please see below for details.

Main comments:

How does the multiscale data assimilation implemented here compare to previous work, such as Li et al (2015) and Tissier et al (2019)? I understand more details should be available in Chamberlain et al (2021a). But since this is not yet published, it would be good to have some more details in the present manuscript.

The magnitude of improvements found with the multiscale data assimilation in BRAN were comparable to those reported in regional and basin scale models by Li et al. and Tissier et al. 2019; this is now added to the introduction. In essence, similar ideas are being applied, albeit with different implementations due to different data assimilation (DA) systems (e.g. 'standard' 3D-VAR and EnOI). Other differences are the domains and time scale, here we apply multiscale DA to a long global-scale ocean reanalysis.
The Chamberlain et al. (2021a, doi:10.1016/j.ocemod.2021.101849) manuscript is accepted and available.
A brief description of the multiscale data assimilation is now a subsection of the 'Ocean data assimilation system.'

The model has the Southern Ocean as one of its regions of interest. The absence of a coupled ice model can have significant impacts on the circulation and water column structure in this part of the world, with potential repercussions to the global deep ocean. Could you please explain the impacts on the model results and how potential problems are minimized?

A paragraph has been added to the introduction to give a better context to the work done.
The development of BRAN is in support of operational ocean forecasting around Australia and has found many other applications across this broad region, now listed in the introduction as well. As such, BRAN has not focused on dynamics close to Antarctica under the influence of processes associated with sea ice at this stage. The Bluelink Project intends to include sea ice in future versions of BRAN to expand the utility of the product.

I am a bit confused about the definition of innovation. To my best knowledge it is defined as the difference between the observations and the model maped to the obs locations. However, all the values presented in Fig. 2 are positive. Does this mean there are model BIASES for all the fields? Or are the innovation and increment defined in a different way?

This definition of innovation is fine, the values plotted in Fig. 2 are the "mean absolute values" which is why they are all positive. There is now a separate paragraph discussing the values of mean biases that may help clarify.
Figure 2 is showing trends in global averages of the absolute innovation values; global averages of mean innovations (~ biases) typically average out close to zero, obscuring significant regional variability, discussed and shown in Chamberlain et al 2021a.

A few minor comments:

Lines 64-65. It looks there is a typo. Please review it.

This sentence has been rewritten to clarify.

Line 97. Typo: "of" should be "on".

Fixed.

Lines 190-191: Comparing the daily average fields to observations excludes a lot of high frequency processes that will be smoothed out. I imagine the observations were also averaged for the comparisons.

Yes and yes.
Rapid, small scale dynamics will be averaged out, there are not enough observations to constrain these features over the global domain at this time.

All available observations from each day are used for these comparisons. Text is modified to help clarify, "we compare daily-averaged reanalysed fields … with daily observations."

Line 215. Could you please explain how the multiscale assimilation eliminated the biases?

In the extra paragraph added discussing biases (Section 3.1), there is a brief explanation of why the multiscale is effective.
"The fine scale corrects mesoscale features (like done in BRAN2016), and the extra coarse step uses an ensemble containing longer length scales and larger localisation (see Appendix), which are more effective at correcting large-scale biases."

Line 216. How to avoid the over-fitting in the EnOI scheme?

We suggest modifying the observation errors assigned, for AVHRR in particular (as described later in this section), which reduces the weighting used to fit to the observations within the DA system when calculating the corrections to the ocean state.

---

## Author Comment (AC4)

**Comment on essd-2021-194**

Anonymous Referee #4

General comment: The technical contents and descriptions are good enough to understand about BRAN2020 reanalysis system. Main conclusion of the paper is about the contribution of multi-scale data assimilation (DA) approach to resolve mesoscale features of ocean conditions. Overall quality of the paper is fairly acceptable. But acceptance decision can be made once a few requests are followed up and discussed.

Thank you for reading the manuscript and the useful comments.

Technical comments:

1. BRAN2020 combines both coarse and fine resolutions in its multi-scale DA approach. One of the main issues is that sea ice model is not included. This naturally leads to what is the benefit of increasing ocean model resolution without considering coupled sea ice ocean modeling system. Concern is that the absence of realistic sea ice condition may deteriorate the analysis result in high latitude areas. Not enough information is provided about the issue along the high latitude areas.

A paragraph has been added to the introduction to give better background and context to our motivations and work presented. The development of BRAN is in support of operational ocean forecasting around Australia and has found many other applications across this broad region, now listed in the introduction as well. As such, BRAN does not focus on dynamics close to Antarctica under the influence of processes associated with sea ice at this stage. To reduce any impact on the properties of deep/dense waters, the model restores temperature and salinity below 2000 m towards climatology. The Bluelink Project intends to include sea ice in future versions of BRAN to increase the utility of the product for research.

2. As relatively fine scale ocean model is used in the multi-scale approach, another natural question is about its benefit on circulation dynamics. Climate index comparisons are described but questions still remain about dynamics: currents, transports, etc.

There is a new subsection added to the paper that compares boundary currents around Australia from the new reanalysis with previous versions. The results are entirely consistent, giving confidence that the multiscale data assimilation has been able to reduce the bias without having an impact on the overall transports and dynamics.

3. Super-observation scheme is used in the reanalysis run from 1993 to 2019. It will be great if authors can provide information about overall computational cost and quantification of observation data quality improvement of the super-obbing in the BRAN2020 system.

The construction of super observations is a preprocessing step, error information is propagated so that the quality of the analysis is the same as if observations were used individually. The preprocessing is relatively quick and cheap, and the difference is the massive saving in DA computation, particularly for SST. For example, where there would be 100M + global observations from a 3-day window, this can be reduced to 2M on the high-resolution grid, or 50k on the coarse grid.

4. In page 10 (line #217), authors used the term "we think" to talk about observation error specification issue. They consider that better analysis result might be obtained if larger error is used for avhrr sst data. To make a conclusive opinion, they have to provide a direct evidence. A small set of analysis experiment might be possible. Without such a direct evidence, it will end up to a simple guessing.

As the text now indicates, in a short test of ~20 cycles the AVHRR SST observation error was increased to 0.3 and showed some improvements in the TEM (0-50m) results (reductions of ~3% in analysis and ~0.5% in background errors) which explain part of the differences seen between BRAN2020 and BRAN2016.

5. The study claims that multi-scale DA approach is beneficial even for non-argo time period (especially before 2000). Most of the comparisons of the study is based on data sets applied in the DA system. OISST and climate index comparisons are provided but additional comparison can be made against another third party reanalysis products.

There has been rewriting in the section comparing values in Tables 2 and 3 and the performance of various DA metrics from BRAN2020

and BRAN2016, including SST. The text now emphasises that the impact of the multiscale DA is primarily in the assimilation of sparse subsurface observations; which is supported by results in Chamberlain et al. (2021a, doi:10.1016/j.ocemod.2021.101849), which is now available and discussed in more detail in the last paragraph of section 3.1.

The improvements in SST (like with SLA) found with BRAN2020 relative to BRAN2016 are smaller than in the subsurface, and are attributed to the new compilations of data that were assimilated into BRAN2020.

Please note that while comparisons are calculated with the same data, forecast/background calculations are made before assimilation.

It is not entirely clear what further comparisons would be most helpful here. In the spirit of the suggestion, a time series was calculated for the RMS of differences between the versions of BRAN and HadISST (see below). Results were consistent with values in the manuscript Tables, the SST background innovations in particular; namely, the RMS of differences between BRAN2020 and HadISST are ~10% less relative to BRAN2016 in the 1990s, whereas the improvement is only ~ a few % in the 2000s and 2010s. While this comparison is useful, it doesn't add new information and is somewhat complicated to explain, for example, why are the RMS values below substantially greater than values in the Tables (related to different processing), hence this is not added to the manuscript.

[Figure]

Figure: RMS of differences between versions of BRAN and monthly HadISST
(http://www.metoffice.gov.uk/hadobs/hadisst)

6. Additional minor editorial comments can be provided once enough technical discussion and feedback is provided.

Thank you again for reviewing the manuscript.
We are prepared to consider further comments if required.